# Pharmacological targeting of caspase-8/ c-FLIP$_L$ heterodimer enhances complex II assembly and elimination of pancreatic cancer cells

Corinna König[1], Nikita V. Ivanisenko[1], Vladimir A. Ivanisenko[2,3], Dagmar Kulms[4,5] & Inna N. Lavrik [1] ✉

Extrinsic apoptotic network is driven by Death Ligand (DL)-mediated activation of procaspase-8. Recently, we have developed the first-in class small molecule, FLIPinB, which specifically targets the key regulator of extrinsic apoptosis, the protein c-FLIP$_L$, in the caspase-8/c-FLIP$_L$ heterodimer. We have shown that FLIPinB enhances DL-induced caspase-8 activity and apoptosis. However, the effects of FLIPinB action in combination with other cell death inducers have only just begun to be elucidated. Here, we show that FLIPinB enhances the cell death in pancreatic cancer cells induced by combinatorial treatment with DL, gemcitabine and Mcl-1 inhibitor S63845. Further, we found that these effects are mediated *via* an increase in the complex II assembly. Collectively, our study shows that targeting the caspase-8/c-FLIP$_L$ heterodimer in combination with the other drugs in pancreatic cancer cells is a promising direction that may provide a basis for further therapeutic strategies.

One of the major characteristics of cancer cells is their resistance towards programmed cell death, in particular to apoptosis. There are two major pathways of apoptosis induction: the extrinsic pathway and the intrinsic/mitochondrial pathway[1,2]. The extrinsic apoptosis pathway is initiated by the activation of death receptors (DRs) such as TRAIL-R1/2 and CD95[3–5]. Upon binding of the cognate death ligands (DLs) TRAIL or CD95L, respectively, the death-inducing signaling complex (DISC) is formed[5–7]. DISC comprises DR, FADD, procaspase-8/10 and c-FLIP$_{L/S/R}$ proteins[8]. DISC serves as a platform for procaspase-8 activation, which takes place upon its oligomerization in the death effector domain (DED)-filaments, initiating the apoptotic destruction of the cell.

c-FLIP$_L$, the long isoform of c-FLIP, is a DED-containing protein and a key regulator of the extrinsic apoptosis pathway[9,10]. The important pro-apoptotic function of c-FLIP$_L$ at the DISC involves its binding to procaspase-8 and subsequent formation of the procaspase-8/c-FLIP$_L$ heterodimer, which is followed by the activation of the caspase-8 enzyme in this heterodimer[11].

In our previous work, using in silico screening and experimental validation, we have developed the first-in class family of lead compounds that specifically bind to c-FLIP$_L$ in the caspase-8/c-FLIP$_L$ heterodimer, stabilize the active center of caspase-8 and increase its activity[12]. These compounds were named FLIPins (FLIP interactors)[12]. The lead compound

FLIPinB increased DL-induced caspase-8 activity as well as promoted cell death in HeLa, Jurkat and Acute Myeloid Leukemia (AML) cell lines. However, the effects of these compounds in other cancer types as well as in combination with conventional anti-cancer drugs are only started to being explored[13,14].

Activation of DRs in combination with inhibition of caspases can also lead to the induction of another pathway of cell death, necroptosis[15,16]. Necroptosis is mediated *via* the assembly of the complex IIb or necrosome[17,18]. The necrosome comprises kinases RIPK1, RIPK3 as well as the pseudokinase MLKL and the DED proteins FADD, procaspase-8/10 and c-FLIP. RIPK1 and RIPK3 are phosphorylated at this macromolecular platform leading to the phosphorylation and activation of MLKL, which drives this type of cell death through membrane pore formation.

Mcl-1 is a member of the anti-apoptotic Bcl-2 family and one of the major regulators of the intrinsic apoptosis pathway[19]. Two isoforms of Mcl-1: Mcl-1$_L$ and Mcl-1$_S$ are the best studied ones[19]. Furthermore, recently, Mcl-1 has received increased attention as being one of the major targets in anti-cancer research[20]. In particular, several compounds were developed that successfully target Mcl-1[21]. In this regard, the small molecule S63845 is one of the leading compounds and was shown to promote apoptotic cell death in several cancer models[21]. Further, it was suggested that targeting Mcl-1 together with c-FLIP proteins comprise an important combination

[1]Translational Inflammation Research, Medical Faculty, Otto von Guericke University (OvGU), Magdeburg, Magdeburg, Germany. [2]Institute of Cytology and Genetics, Novosibirsk, Russia. [3]State Novosibirsk University, Novosibirsk, Russia. [4]Experimental Dermatology, Department of Dermatology, TU-Dresden, Dresden, Germany. [5]National Center for Tumor Diseases, TU-Dresden, Dresden, Germany. ✉e-mail: inna.lavrik@med.ovgu.de

for interfering with the apoptotic network in cancer cells[22]. Hence, investigation of potential co-targeting of Mcl-1 and c-FLIP presents an important direction in anti-cancer studies.

Pancreatic ductal adenocarcinoma (PDAC) is the fourth leading cause of cancer-related deaths worldwide and has the highest mortality rate compared to other solid tumors[23]. Gemcitabine is a first-line drug that is used in the conventional chemotherapy of PDAC, both as monotherapy as well as in combination with other chemotherapeutics[24,25]. However, severe toxic effects of these therapies are well-documented along with a low therapeutic benefit for patients. Hence, the development of more specific therapeutic approaches for PDAC is urgently required.

One of the promising directions is specifically targeting the components of the PDAC cell death network in combination with the first-line chemotherapeutics used for the combinatorial treatment. In this respect, an emerging direction for therapy is combinatorial treatment based on TRAIL and its recently designed analogs[4,26,27]. Moreover, the combination of TRAIL analogs and gemcitabine was investigated in clinical trials[28]. TRAIL, as DL inducing extrinsic apoptosis, is highly attractive for anti-cancer therapies since it was reported to selectively kill tumor cells in vitro and in vivo[4]. In our previous studies, we have investigated the effects of gemcitabine in combination with DLs on the cell death network in pancreatic cancer cells[29]. Our former study has shown that co-treatment with DL/gemcitabine leads to the induction of two cell death programmes in pancreatic cancer cells: apoptosis and necroptosis[29]. Apoptosis was mediated via induction of both intrinsic and extrinsic pathways. Moreover, this study has suggested that c-FLIP$_L$ and Mcl-1 might present valuable targets in the cell death network induced by DL/gemcitabine treatment, which is in accordance with other reports[22,29]. Finally, there is evidence that c-FLIP proteins may represent a promising target in PDAC, suggesting the importance of investigating pharmacological targeting of c-FLIP in this type of cancer[30].

In the current study, we investigated the effects of FLIPinB, the first-in-class compound targeting c-FLIP$_L$, on DL/gemcitabine-induced cell death networks in pancreatic cancer cells. In particular, we considered combinatorial treatments based on DL/gemcitabine together with targeting of c-FLIP$_L$ with FLIPinB and of Mcl-1 with S63845. This combination was efficient in the elimination of pancreatic cancer cells, which was mediated via an increase of the complex II formation.

## Results

### DL-induced loss of cell viability is enhanced by gemcitabine co-treatment

To investigate the effects of FLIPinB on DL/gemcitabine treatment, first we checked the expression of the key DR signaling components in pancreatic cancer cell lines: SUIT-020, MiaPaca2, and Panc89 comparing them to colon cancer HT29 cells. The latter was used as a well-characterized model cell line for both apoptosis and necroptosis induction. SUIT-020, MiaPaca2 and Panc89 cell lines were characterized by the expression of all major components of both extrinsic and intrinsic cell death pathways including TRAIL-R1/2, CD95, procaspase-8a/b, procaspase-10a/d, c-FLIP$_{L/S}$, FADD, procaspase-3, XIAP, RIPK1, RIPK3, MLKL and Bcl-2 family members (Fig. 1a, Supplementary Fig. 1a–d). The expression of most components of the DR network was similar among the three cell lines, but there were some differences. XIAP proteins were higher expressed in SUIT-020 compared to MiaPaca2 and Panc89 cells, suggesting the potential resistance of this cell line to DL stimulation[31–33]. Consistent with their potential resistant phenotype, SUIT-020 cells were also characterized by the lower expression of pro-apoptotic Bax and Bak as well as the higher expression of anti-apoptotic Bcl-xL. Mcl-1$_L$ and Bcl-2 have been expressed to a higher extent in MiaPaca2 cells compared to the other two cell lines (Fig. 1a). CD95 had the strongest expression in SUIT-020 compared to Panc89 and MiaPaca2 cells (Fig. 1a, Supplementary Fig. 1b, d). The expression of the core components of the necroptotic network in pancreatic cancer cell lines was compared to colon cancer HT29 cells, which, as above mentioned, are a common model for cells that can undergo necroptosis. MiaPaca2 cells did not express RIPK3, while Panc89 cells

showed the higher expression of RIPK1, RIPK3, and MLKL compared to SUIT-020 cells.

In line with the higher expression of XIAP in SUIT-020 cells, MiaPaca2 and Panc89 have shown more viability loss upon TRAIL treatment compared to SUIT-020 cells (Supplementary Fig. 1e–g). Accordingly, these three cell lines were taken for analyzing the effects of FLIPinB on DL/gemcitabine networks. Specifically, MiaPaca2 and Panc89 cells were considered as DL-sensitive cell lines, while SUIT-020 cells were considered as a model of a resistant cell line.

We have aimed at selecting the lowest concentrations of the cell death-inducing stimuli which when applied alone do not induce any cytotoxic effects but work in combination to eliminate pancreatic cancer cells. In this way, we aimed to mimic the effects for reducing the possible side effects of combinatorial treatment using pancreatic cancer cell lines as a model system. First, our goal was on selecting the lowest concentration of gemcitabine for DL/gemcitabine treatment. In line with the high expression of XIAPs, SUIT-020 cells were largely resistant towards CD95L and TRAIL treatments; however, the efficiency of TRAIL- and CD95L-induced viability loss of SUIT-020 cells was largely increased upon gemcitabine co-treatment (Fig. 1b, c). The sensitization with gemcitabine was dose-dependent and effects were already detected upon pre-treatment with 10 ng/ml of gemcitabine for 24 h and subsequent co-stimulation with DL (Fig. 1b, c). Sensitization towards DL stimulation via gemcitabine co-treatment was also observed in MiaPaca2 cells (Fig. 1d, e). Moreover, MiaPaca2 cells were less sensitive towards CD95L compared to TRAIL treatment (Fig. 1e), which was consistent with the low expression levels of CD95 in these cells (Fig. 1a). Panc89 cells were more sensitive towards both CD95L and TRAIL treatments compared to MiaPaca2 and SUIT-020, as well as Panc89 cells can be efficiently sensitized towards DL stimulation by gemcitabine (Fig. 1f, g). These experiments show that gemcitabine enhances cell viability loss upon TRAIL and CD95L co-treatment in SUIT-020, MiaPaca2, and Panc89 cells already starting from a concentration of 10 ng/ml gemcitabine. Hence, the concentration of gemcitabine equal to 10 ng/ml was selected for the further analysis.

DL/gemcitabine-induced cell viability loss upon co-treatment with 10 ng/ml of gemcitabine was not accompanied by an increase of procaspase-3 and PARP1 cleavage products as was observed in SUIT-020 cells (Supplementary Fig. 2a–d). This pattern of caspase-3 cleavage fits well to the previous observations of caspase processing upon the stimulation with threshold concentrations of DLs or other death stimuli[34].

### FLIPinB compound shows activity in MiaPaca2 and Panc89 cells

Previously, we showed that the first-in-class small molecule targeting c-FLIP$_L$, FLIPinB (Fig. 2a, b), enhances DL-induced caspase-8 activity and apoptosis by stabilizing the active center of the caspase-8/c-FLIP$_L$ heterodimer[12]. The efficiency of FLIPinB action depends on the expression levels of c-FLIP$_L$ in the particular cell type and its ratio to procaspase-8 as well as the resulting quantity of procaspase-8/c-FLIP$_L$ heterodimers formed at the DISC. Further, the amount of DRs and assembled active complexes play an important role[12,35]. We constructed a mathematical model to predict the optimal ratio of c-FLIP$_L$ and procaspase-8 levels in the DED filament leading to the highest sensitivity to FLIPinB/CD95L co-treatment[35]. According to the model predictions, the optimal ratio was in the range of 2:1 to 3:1 of procaspase-8:c-FLIP$_L$. Interestingly, the amounts of procaspase-8 and c-FLIP$_L$ were similar in SUIT-020 cells compared to MiaPaca2 cells (Fig. 1a). However, the expression levels of XIAP were higher in SUIT-020 cells (Fig. 1a), which should result in the lower effector caspase activity in these cells leading to the lower rate of cell viability loss even upon co-administration of FLIPinB acting on the initiator caspase-8[32]. In Panc89 cells, c-FLIP$_L$ was higher expressed compared to the other two cell lines (Fig. 1a), which should result in the higher number of heterodimers formed and more efficient FLIPinB action. Therefore, it might be suggested that FLIPinB has stronger effects on TRAIL-induced cell viability loss in MiaPaca2 and Panc89 cells compared to SUIT-020 cells[36].

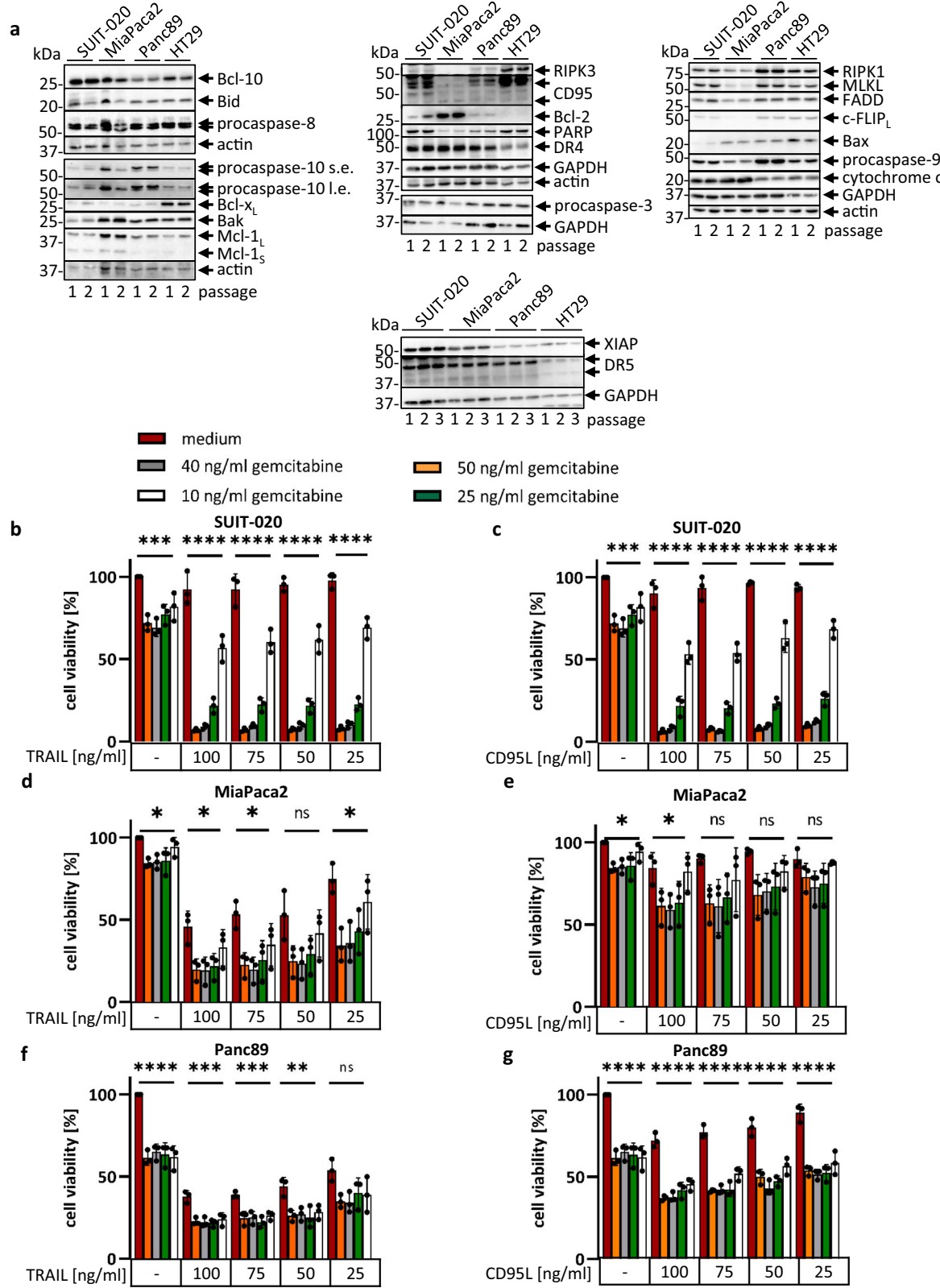

**Fig. 1 | TRAIL/gemcitabine and CD95L/gemcitabine co-treatments induce loss of cell viability. a** Total cell lysates of SUIT-020, MiaPaca2, Panc89 and HT29 cells from two or three different passages were compared by Western Blot. Protein expression was analyzed using the indicated antibodies. Actin or GAPDH served as loading controls. SUIT-020 cells (**b, c**), MiaPaca2 cells (**d, e**), and Panc89 cells (**f, g**) were pretreated for 24 h with the indicated concentrations of gemcitabine and afterwards treated for 22 h with TRAIL (**b, d, f**) or CD95L (**c, e, g**). ATP content was measured using the Cell Titer-Glo®-Luminescent Cell Viability Assay. Mean and standard deviation (SD) from three independent experiments are shown. The error bars indicate the mean ± SD. The results presented in the panels (**b, c**), (**d, e**), and (**f, g**) were obtained in the same experiment, e.g., using one 96-well plate. For the statistical analysis One-way ANOVA tests were used to compare the group of conditions for each of TRAIL or CD95L treatments. Each group comprised the treatment with one concentration of TRAIL/CD95L. The following values were used: ****$p < 0.0001$; ***$p < 0.001$; **$p < 0.01$; *$p < 0.05$; ns not significant. Abbreviations: s.e. short exposure, l.e. long exposure.

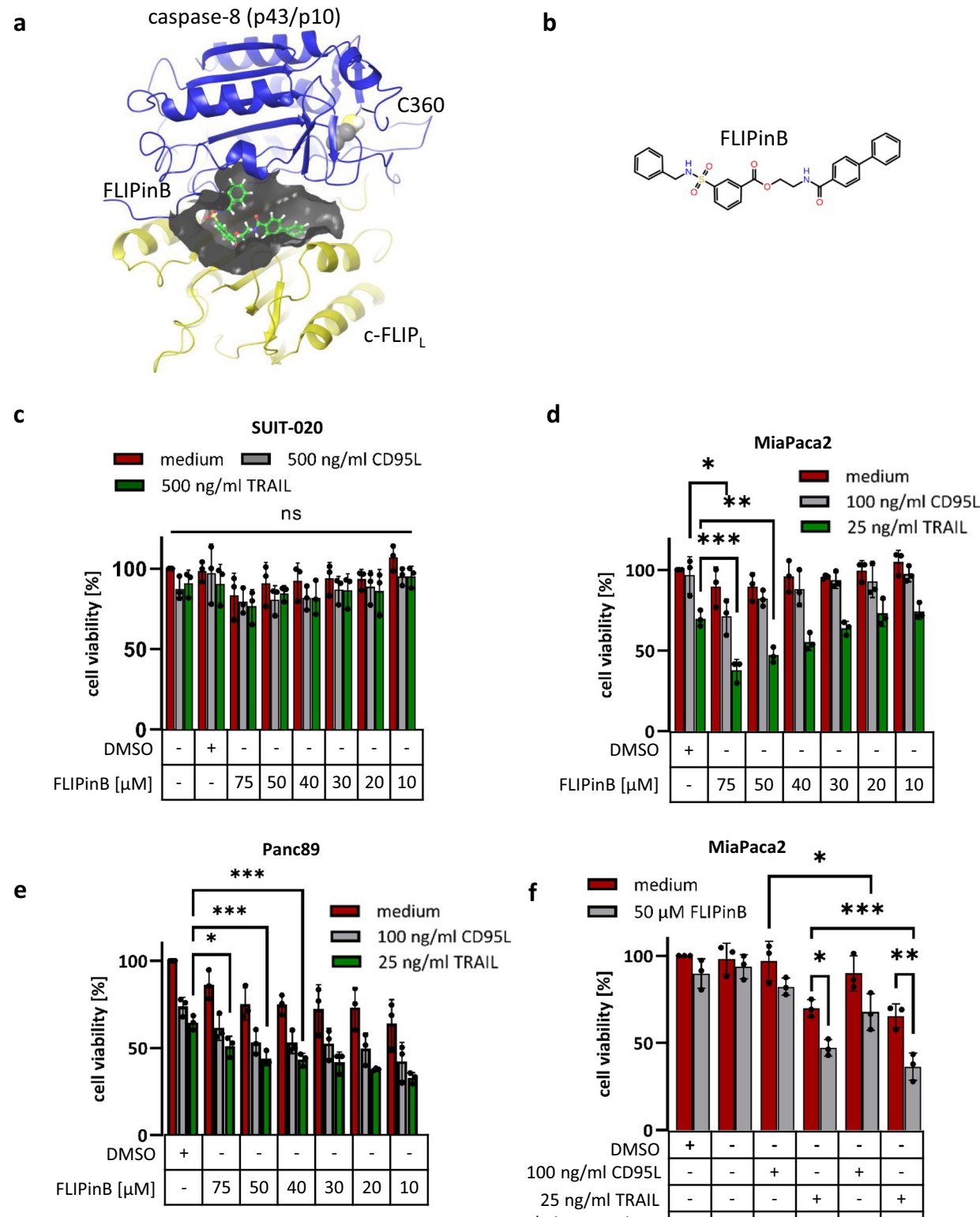

**Fig. 2 | FLIPinB compound shows activity on MiaPaca2 and Panc89 cells.**
**a** Scheme of the caspase-8/c-FLIP$_L$ heterodimer with caspase-8 (p43/p10) in blue and c-FLIP$_L$ in yellow. The active site cysteine C360 is shown in yellow. FLIPinB is shown in green as bound to the interface of the caspase-8/c-FLIP$_L$ heterodimer.
**b** The structural formula of FLIPinB is shown. SUIT-020 (**c**), MiaPaca2 (**d**) or Panc89 cells (**e**) were pretreated for 2 h with the indicated concentrations of FLIPinB. Subsequently, cells were treated for 22 h with 500 ng/ml TRAIL or 500 ng/ml CD95L (**c**) or 25 ng/ml TRAIL or 100 ng/ml CD95L (**d**, **e**). **f** MiaPaca2 cells were treated for

24 h with 10 ng/ml gemcitabine. Afterwards the cells were stimulated for 2 h with 50 μM FLIPinB and subsequently for 22 h with 25 ng/ml TRAIL or 100 ng/ml CD95L. ATP content was measured using the Cell Titer-Glo®-Luminescent Cell Viability Assay. Mean and SD from three independent experiments are shown. The error bars indicate the mean ± SD. For statistical analysis One-way ANOVA (**c**) to compare a group or One-way ANOVA with Tukey post hoc tests (**d**–**f**) were used to compare two conditions. The following values were used: ****$p < 0.0001$; ***$p < 0.001$; **$p < 0.01$; *$p < 0.05$; ns not significant.

FLIPinB has enhanced TRAIL- or TRAIL/gemcitabine-induced cell viability loss of MiaPaca2 and DL-induced cell viability loss of Panc89 cells, while almost no effects were observed in SUIT-020 cells in combination with DL or DL/gemcitabine treatments (Fig. 2c–f; Supplementary Fig. 2e, f). This indicates that SUIT-020 cells were rather resistant towards DL treatment, which was consistent with the suggestions made above based on the expression levels of the core components of extrinsic cell death network (Fig. 1a). However, the effects of FLIPinB were also not very strong in MiaPaca2 and Panc89 cells. This, in turn, suggested further exploring whether the effects of FLIPinB might be enhanced in combination with inhibitors of other key targets of the cell death pathway at the level of mitochondria such as Mcl-1.

## Mcl-1 inhibitor S63845 has different activity in three pancreatic cancer cell lines

S63845 is a well-established inhibitor of Mcl-1 that triggers apoptosis in a number of cancer cells[21]. Interestingly, treatment of SUIT-020 cells with S63845, in a range of conventionally used concentrations from 0.5 to 10 μM, did not lead to a loss of cell viability within 24 h (Fig. 3a). In contrast, MiaPaca2 and Panc89 cells were more sensitive towards S63845 treatment (Fig. 3b, c). The sensitivity of these cells towards Mcl-1 inhibitor was consistent with the levels of Mcl-1 expression in these cells, being the lowest in SUIT-020 cells (Fig. 1a). Moreover, the high levels of anti-apoptotic XIAP and Bcl-xL in SUIT-020 cells may also contribute to their lower sensitivity towards S63845. The low sensitivity of SUIT-020 cells towards S63845 treatment was also observed in Western Blot analysis of caspase cleavage (Supplementary Fig. 3a). Indeed, neither processing of procaspases-8, -10 and -3, nor cleavage of their substrates c-FLIP or PARP1 were detected upon treatment with up to 20 μM of S63845 for 6 h. This was in contrast to MiaPaca2 and Panc89 cells, in which procaspase-8 and -3 processing and cleavage of the caspase-3 substrate PARP1 were observed using even the lower concentrations of S63845 (Supplementary Fig. 3b, c). The upregulation of the short isoform of Mcl-1, Mcl-1$_S$, in these experiments was observed in a concentration-dependent manner upon S63845 administration in accordance with previous reports[19].

Next, we examined the concentrations of S63845, which can be used for sensitization of these three cell lines towards DL/gemcitabine/FLIPinB treatment. For combinatorial treatments, we have selected low concentrations of gemcitabine and S63845, since as highlighted, our aim was to use low or threshold doses of stimuli that work in combination to eliminate the cancer cells. Treatment with 10 ng/ml gemcitabine was used for all cell lines as discussed above. To select the concentration of S63845, this inhibitor was added to the cells in a dose-dependent manner in combination with DL/gemcitabine. The threshold concentration of S63845 was then selected as the concentration in the range between the one that caused the decrease of cell viability and the one that did not influence the cell viability loss. Specifically, in SUIT-020 cells, 20 μM S63845 in combination with DL/gemcitabine induced loss of cell viability, while 10 μM did not (Fig. 3d, Supplementary Fig. 3d). Therefore, 10 μM S63845 was chosen as the concentration for treatment of SUIT-020 cells (Fig. 3d, Supplementary Fig. 3d). For MiaPaca2 cells, no difference was observed for the S63845 concentration range of 0.1–2.5 μM in DL/gemcitabine treatment (Fig. 3e, Supplementary Fig. 3e), while 5 μM caused a stronger decrease in cell viability loss compared to 2.5 μM upon administration alone (Fig. 3b). Therefore, the concentration of S63845 equal to 2.5 μM was selected for MiaPaca2 cells. For Panc89 cells, the administration of 0.5 μM S63845 already caused almost complete loss of cell viability in combination with either DL or DL/gemcitabine treatments (Fig. 3f). Hence, a further decrease in the concentration of S63845 for combinatorial treatments of Panc89 cells was required based on experiments with the administration of S63845 alone (Fig. 3c). These experiments have demonstrated that S63845 has the most prominent effects upon combined treatment of Panc89 cells.

## FLIPinB sensitizes pancreatic cancer cells towards DL/gemcitabine/S63845 treatment

Next, the effects of FLIPinB on DL/gemcitabine/S63845 co-stimulation in SUIT-020, MiaPaca2, and Panc89 cells were examined. The simultaneous

administration of all four agents (DL/gemcitabine/FLIPinB/S63845) led to more efficient cell viability loss than upon double (DL/gemcitabine) or triple (DL/gemcitabine/S63845) treatments (Fig. 4a–d, Supplementary Fig. 3f, g). This was observed in all three cell lines for both TRAIL (Fig. 4a, c, d, Supplementary Fig. 3f, g) and CD95L stimulations (Fig. 4b, c, d, Supplementary Fig. 3f, g). MiaPaca2 cells were more sensitive to DL/gemcitabine/FLIPinB treatment compared to SUIT-020 cells and the addition of S63845 enhanced the effects. FLIPinB also efficiently sensitized Panc89 cells towards DL/gemcitabine/S63845 treatments (Fig. 4d). Furthermore, the sensitization effects in Panc89 cells were observed not only using viability assays but also the cytotoxicity assays based on LDH release (Fig. 4e).

Importantly, the increase in cell viability loss upon quadrupole DL/gemcitabine/S63845/FLIPinB stimulation compared to double (DL/gemcitabine) or triple (DL/gemcitabine/S63845 and DL/gemcitabine/FLIPin) treatments was also observed in Panc89 cells growing in 3D as tumor-mimicking spheroids (Fig. 5a–c). Importantly, in these experiments, the quadrupole treatment was the most efficient one for both CD95L and TRAIL treatments.

Since we aimed at using the lowest concentrations of death-inducing agents that, when applied alone, might even not induce cell death but work in combination to enhance the elimination of the cells, we checked whether the administration of gemcitabine, S63845 or FLIPinB to SUIT-020 cells without addition of DLs might lead to the appearance of any apoptotic markers such as cleavage products of caspase-3 or PARP1 (Supplementary Fig. 3h). The appearance of caspase cleavage products was not observed indicating that these concentrations of inhibitors do not cause any apoptosis induction or cytotoxic effects without DL co-treatment (Supplementary Fig. 3h).

The major attention in targeting apoptotic network belongs to BH3 mimetics. In this regard, we also investigated the effects of co-treatment with DL/gemcitabine/FLIPinB and ABT-263/navitoclax. ABT-263 is an inhibitor of two other key anti-apoptotic Bcl-2 family members: Bcl-2 and Bcl-x$_L$[37]. ABT-263 was very efficient in sensitizing the most resistant SUIT-020 cells towards cell death. Namely, triple co-treatment with DL/gemcitabine/ABT-263 led to a strong dose-dependent loss of cell viability even upon low concentrations of ABT-263 (Supplementary Fig. 4a, b). In line with these results, almost no difference was observed in co-treatment with DL/gemcitabine/FLIPinB/ABT-263 compared to DL/gemcitabine/ABT-263 (Supplementary Fig. 4c–f). This indicated that DL/gemcitabine/ABT-263 co-treatment had already a high efficiency without further requirement for sensitization by FLIPinB. Subsequently, we did not further consider this direction since the goal of the study was to investigate the effects of FLIPinB in combinatorial treatments on pancreatic cancer cells.

The treatment of cancer cells with gemcitabine has been reported to induce upregulation of the DL and DR level, in particular, TRAIL-R2 and TNFα[38–41]. This, in turn, can lead to an increase of cell viability loss upon DL/gemcitabine/FLIPinB/S63845. Accordingly, the changes in the expression of TRAIL-Rs upon double (DL/gemcitabine), triple (DL/gemcitabine/S63845) and quadrupole (DL/gemcitabine/FLIPinB/S63845) treatments were monitored. However, no upregulation of TRAIL-Rs upon DL/gemcitabine/FLIPinB/S63845 was observed compared to double and triple treatments (Supplementary Fig. 5a). TNF-α production was slightly increased upon triple and quadrupole co-treatments, but the levels of TNF-α increase were rather low (Fig. 5d). Therefore, we suggested that upregulation of TNF-α also did not contribute to the increase in cell viability loss. These experiments ruled out the involvement of DL/DR upregulation upon this co-stimulation and suggested further exploring the molecular network of DL/gemcitabine/FLIPinB/S63845 co-treatment.

## DL/gemcitabine/FLIPinB/S63845 treatment of pancreatic cancer cells leads to apoptosis and necroptosis induction

In the previous work, it has been reported that DL/gemcitabine induces both apoptotic and necroptotic cell death programmes in pancreatic cancer cells[29]. Hence, next we have checked the induction of these two cell death pathways upon combinatorial treatment with DL, gemcitabine, FLIPinB

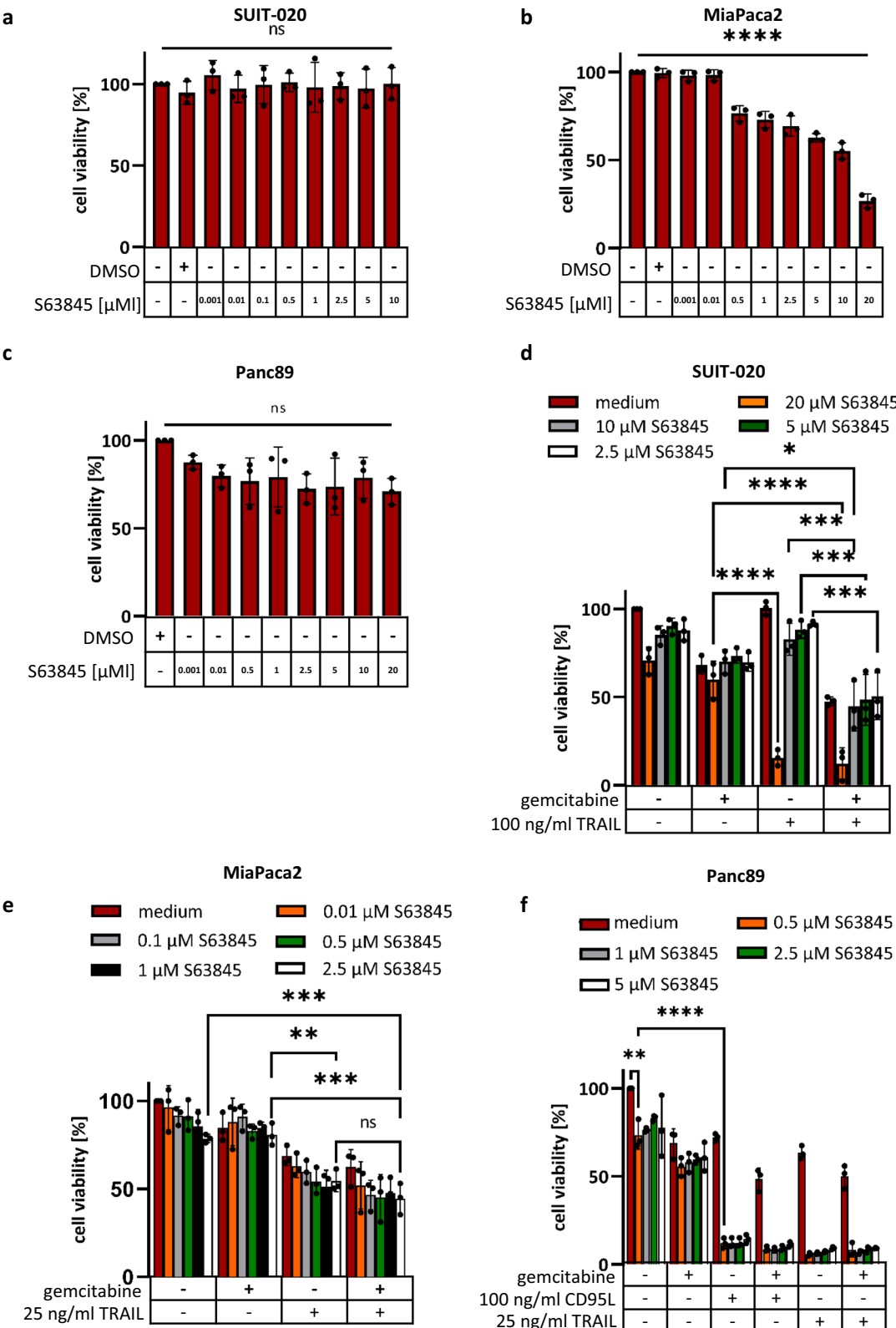

**Fig. 3 | Mcl-1 inhibitor S63845 has different activity in three pancreatic cancer cell lines.** SUIT-020 cells (**a**), MiaPaca2 cells (**b**) or Panc89 cells (**c**) were treated with the indicated concentration of S63845 for 24 h. SUIT-020 cells (**d**), MiaPaca2 cells (**e**) or Panc89 cells (**f**) were treated with 10 ng/ml gemcitabine for 24 h. This was followed by S63845 treatment for 2 h with the indicated concentrations and subsequent DL stimulation for 22 h. ATP content was measured using the Cell Titer-Glo®-

Luminescent Cell Viability Assay. Mean and SD from three independent experiments are shown. The error bars indicate the mean ± SD. For statistical analysis One-way ANOVA (**a–c**) to compare a group or One-way ANOVA with Tukey post hoc tests (**d–f**) were used to compare two conditions. The following values were used: ****$p < 0.0001$; ***$p < 0.001$; **$p < 0.01$; *$p < 0.05$; ns not significant.

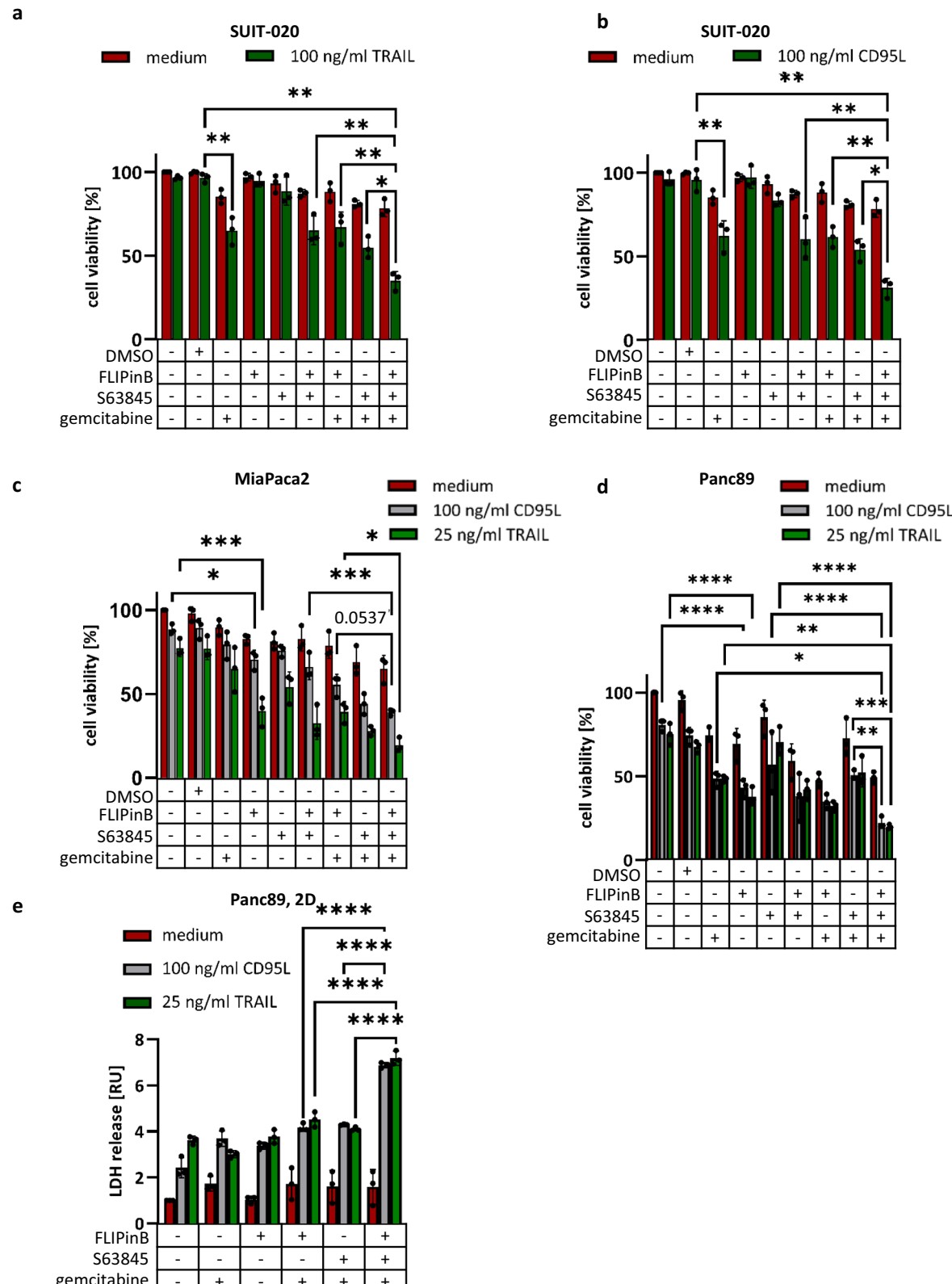

**Fig. 4 | FLIPinB sensitizes pancreatic cancer cells towards DL/gemcitabine/ S63845 treatment.** SUIT-020 cells (**a**, **b**), MiaPaca2 cells (**c**) or Panc89 cells (**d**) were pretreated for 24 h with 10 ng/ml gemcitabine, followed by 2 h treatment with 10 (**a**, **b**), 2.5 (**c**) or 0.005 μM (**d**) S63845 and 20 (**a**, **b**, **d**) or 50 μM (**c**) FLIPinB. Afterwards, the cells were stimulated with 100 ng/ml (**a**) or 25 ng/ml (**c**, **d**) TRAIL or 100 ng/ml CD95L (**b**–**d**) for 22 h. ATP was measured using the Cell Titer-Glo®- Luminescent Cell Viability Assay. **d** Panc89 cells were pretreated for 24 h with 10 ng/ ml gemcitabine and subsequently treated for 2 h with 20 μM FLIPinB and 0.005 μM S63845. Afterwards cells were stimulated for 22 h with 100 ng/ml CD95L or 25 ng/ ml TRAIL. LDH release was measured using the LDH-Glo™ Cytotoxicity Assay. Mean and SD from three independent experiments are shown. The error bars indicate the mean ± SD. For statistical analysis One-way ANOVA with Tukey post hoc tests were used to compare two conditions. The following values were used: ****$p < 0.0001$; ***$p < 0.001$; **$p < 0.01$; *$p < 0.05$; ns not significant.

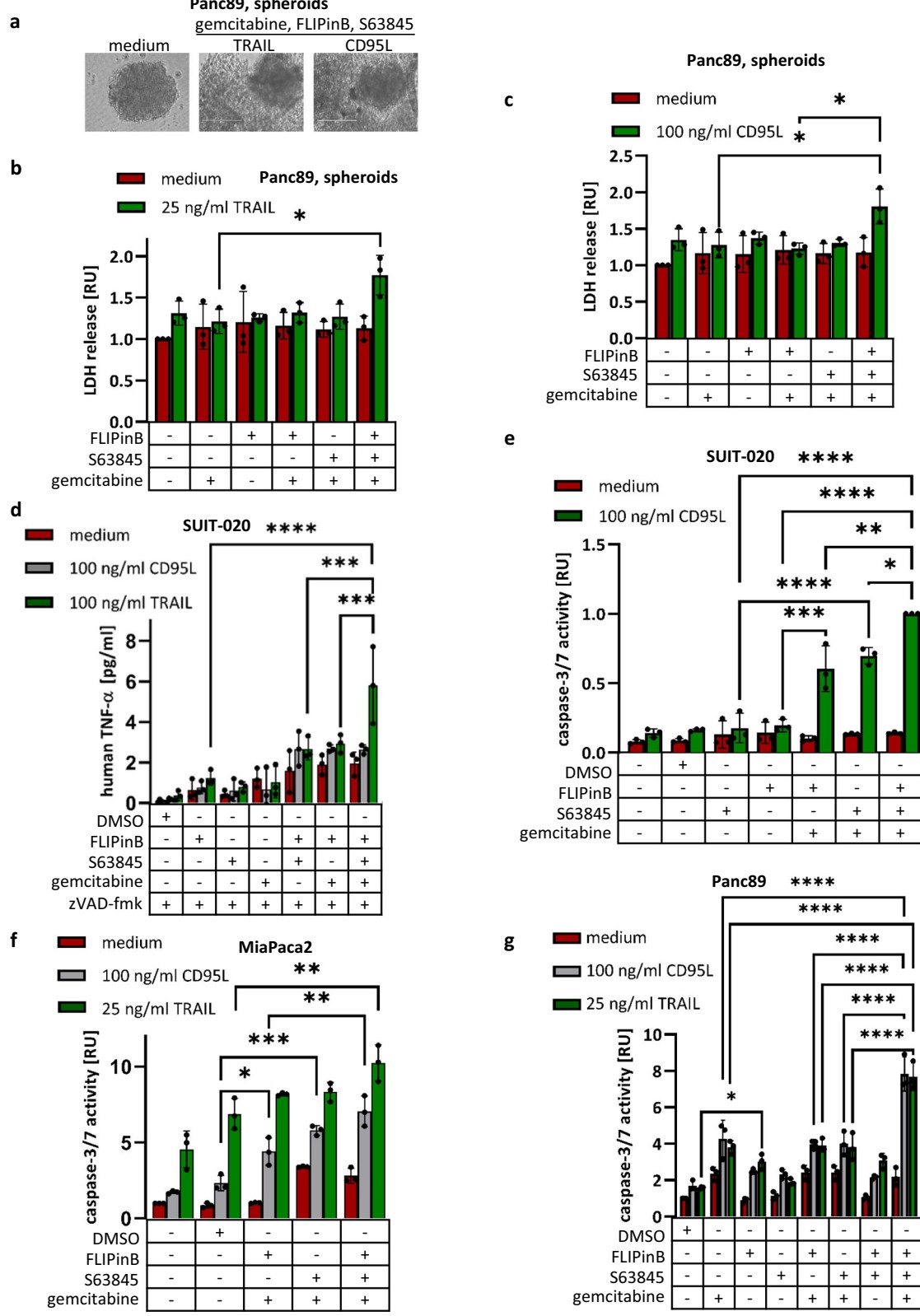

and S63845. A significant increase in caspase-3/7 activity has been detected upon DL/gemcitabine/FLIPinB/S63845 treatments compared to triple treatments in SUIT-020, MiaPaca2, and Panc89 cells (Fig. 5e–g). These results were in accordance with the results of the viability assays and demonstrated that administration of all four stimuli leads to the strongest increase in caspase-3/7 activity in all three cell lines.

To explore the effects on cell death, Imaging Flow Cytometry approach reported by us previously was used that is based on Annexin-V and Propidium Iodide (PI) staining[42]. The amount of dead cells was higher upon DL/gemcitabine/FLIPinB/S63845 treatment of SUIT-020 and Panc89 cells compared to corresponding double and triple treatments (Fig. 6; Supplementary Fig. 5b–d). This was in line with the results of the cell viability,

**Fig. 5 | DL/gemcitabine/FLIPinB/S63845 quadrupole treatment shows the strongest LDH release and effector caspase activation. a** Representative pictures of Panc89 spheroids grown in 3D-culture are shown. **b, c** Panc89 spheroids were pretreated for 24 h with 10 ng/ml gemcitabine and subsequently treated for 2 h with 20 μM FLIPinB and 0.005 μM S63845. Afterwards cells were stimulated for 22 h with 25 ng/ml TRAIL (**b**) or 100 ng/ml CD95L (**c**). LDH release was measured using the LDH-Glo™ Cytotoxicity Assay. **d** SUIT-020 cells were stimulated for 24 h with 10 ng/ml gemcitabine followed by treatment for 2 h with 20 μM FLIPinB and 10 μM S63845. 1 h before CD95L and TRAIL treatment 50 μM zVAD-fmk was added.

CD95L and TRAIL treatment was performed for 22 h. Human TNF-α was measured using ELISA MAX™ Standard Set (BioLegend). SUIT-020 cells (**e**), MiaPaca2 cells (**f**) or Panc89 cells (**g**) were pretreated for 24 h with 10 ng/ml gemcitabine and subsequently treated for 2 h with 20 (**e, g**) or 50 (**f**) μM FLIPinB and 10 (**e**), 2.5 (**f**) or 0.005 (**g**) μM S63845. Afterwards cells were stimulated for 6 h (**e**) or 3 h (**f, g**) with 100 ng/ml CD95L (**e–g**) or 25 ng/ml TRAIL (**f, g**). Caspase-3/7 activity was measured using the Caspase-Glo®3/7 Assay. Mean and SD from three independent experiments are shown. The error bars indicate the mean ± SD. The following values were used: ****$p < 0.0001$; ***$p < 0.001$; **$p < 0.01$; *$p < 0.05$; ns not significant.

cytotoxicity and caspase activity assays and shows that administration of all four stimuli leads to the strongest increase in cell death.

To test whether this combinatorial treatment induces cell death in non-cancerous cells, the normal human fibroblasts were used. Importantly, no increase in cell death was observed for normal human fibroblasts treated with TRAIL/gemcitabine/FLIPinB/S63845 indicating the higher efficiency of this combination for cancer cells (Supplementary Fig. 6).

Imaging Flow Cytometry also allows distinguishing between apoptotic and necrotic or necroptotic cells using its imaging functions[42]. Importantly, when analyzing PI-positive cells, along with late apoptotic cells that are characterized by shrunken nuclei, which can be observed as "sharp red spots" (Fig. 6e, f, right panels), a small number of cells with the typical morphology of necrosis or necroptosis were detected in both cell lines (Fig. 6e, f, middle panels). In particular, the necroptotic or necrotic features include the swollen nucleus and an increase in cell volume, as observed by a more diffuse red color spread over the whole cell.

The appearance of cells with both apoptotic and necrotic morphology upon DL/gemcitabine/FLIPinB/S63845 treatment was also observed under the microscope, though the population of the necrotic cells was rather low (Fig. 7a, middle panel). The addition of pan-caspase inhibitor zVAD-fmk increased the amount of cells with necrotic morphology in both SUIT-020 and Panc89 cells indicating necroptosis induction upon zVAD-fmk addition (Fig. 7a, right panel). Furthermore, we quantified the cells with necrotic morphology from imaging flow cytometry experiments (Fig. 7b, c; Supplementary Fig. 7). For this quantification, we used an approach that was developed previously and is based on gating based on the size of the nucleus of apoptotic *versus* necrotic cells, e.g., distinguishing between small *versus* large nuclei using special features of imaging flow cytometry[42] (Supplementary Fig. 7e, f). For this analysis we used only the population of PI-positive cells, which included PI-positive ones for SUIT-020 and PI/annexin V-positive cells for Panc89. This analysis has shown that the necroptotic cells comprise about 60% of population upon DL/gemcitabine/FLIPinB/S63845/zVAD-fmk treatment (Fig. 7b, Supplementary Fig. 7e, f). This demonstrates that cells undergo necroptosis upon DL/gemcitabine/FLIPinB/S63845 treatment in combination with zVAD-fmk administration.

The induction of necroptosis was further supported by detecting phosphorylation of RIPK1, RIPK3 and MLKL by Western Blot in SUIT-020 and Panc89 cells (Fig. 7d–f). In line with the cell death analysis, CD95L/gemcitabine/FLIPinB/S63845/zVAD-fmk treatment led to the appearance of the necroptotic markers pRIPK1, pRIPK3 and pMLKL (Fig. 7d–f). Moreover, already the double and triple treatments of Panc89 cells in combination with zVAD-fmk led to the appearance of pRIPK1 and pMLKL signals, the latter was enhanced upon quadrupole treatment. Gemcitabine administration alone did not lead to phosphorylation of MLKL indicating DL-dependent appearance of necroptotic markers in these cells (Fig. 7d, e). To validate the induction of the necroptotic pathway we have used the inhibitors of necroptosis, GSK872 as well as Nec-1s and measured the cell viability loss. The treatment with 10 μM GSK872 in combination with DL/gemcitabine/FLIPinB/S63845/zVAD-fmk led to the rescue of the viability of SUIT-020 cells supporting necroptosis induction (Supplementary Fig. 8a). The effects of Nec-1s on rescue of DL/gemcitabine/FLIPinB/S63845/zVAD-fmk-treated cells were not detected likely due the lower efficiency of Nec-1s (Supplementary Fig. 8b–e). MiaPaca2 cells do not have RIPK3, hence, they cannot undergo necroptosis and were not sensitive to Nec-1s treatment under these conditions (Supplementary Fig. 8f, g). Accordingly, adding

zVAD-fmk to any of the treatments in MiaPaca2 cells resulted in the inhibition of their viability loss pointing out that the treatment with DL/gemcitabine/FLIPinB/S63845 leads to caspase-dependent cell death in MiaPaca2 cells.

## CD95L/gemcitabine/FLIPinB/S63845 treatment leads to the increased assembly of complex II

Next, we aimed to further explore the cell death network upon DL/gemcitabine/FLIPinB/S63845 treatment. The analysis of procaspase-8 and procaspase-3 cleavage by Western Blot showed an increase of caspase processing upon DL/gemcitabine/FLIPinB/S63845 treatment compared to all other conditions, both for TRAIL and CD95L in SUIT-020, MiaPaca2 and Panc89 cells (Fig. 8). The increase in procaspase-8 and -3 processing as manifested by the appearance of p18 and p19/p17 cleavage products, respectively, was stronger in MiaPaca2 and Panc89 (Fig. 8c–e) compared to SUIT-020 cells (Fig. 8a, b). In the latter case only very minor increase in p18-caspase-8, p19/p17-caspase-3, and PARP1 cleavage were observed.

To get more insight into caspase-8 activation upon DL/gemcitabine/FLIPinB/S63845 treatment, the DISC assembly was analyzed[6]. DISC-immunoprecipitation (IP) analysis revealed a small increase in procaspase-8 processing upon DL/gemcitabine/FLIPinB compared to DL/gemcitabine/S63845 treatments in SUIT-020 cells, which is in accordance with the mechanism of FLIPinB action. However, no increase in DISC assembly upon quadrupole treatment was observed (Supplementary Fig. 9a). The analysis of the DISC complexes immunoprecipitated from High Molecular Weight fractions (HMW) of CD95L/gemcitabine/FLIPinB/S63845-treated *versus* CD95L/gemcitabine/S63845-treated cells also did not reveal strong differences (Supplementary Fig. 9b).

Upon genotoxic stress caspase-8 can be activated at the complex IIa or RIPoptosome, which is an intracellular platform of caspase-8 activation[43]. The increase in the complex II assembly as well as caspase-8 processing at this complex were observed upon administration of FLIPinB to DL/gemcitabine as well as all four stimuli in MiaPaca2 and Panc89 cells (Fig. 9a, b; Supplementary Fig. 9c). In particular, the increase in the amounts of FADD as well as procaspase-8 and c-FLIP cleavage products: p43/p41-caspase-8, p30-caspase-8, p18-caspase-8 and p43-FLIP, respectively, was detected (Fig. 9a, b, Supplementary Fig. 9c). In line with these observations, an increase in the cleavage of effector caspases, PARP1 and Bid was monitored in total cellular lysates upon administration of all four stimuli (Fig. 9a, b). This points out that FLIPinB administration leads to the increase of complex II, which is further enhanced by the addition of S63845.

The small increase in the amount of complex II was also observed in SUIT-020 cells upon CD95L/gemcitabine/FLIPinB/S6384 co-treatment compared to three other conditions (Supplementary Figs. 9d, 10a). Accordingly, a slightly elevated level of caspase-8 and c-FLIP cleavage products was detected, which indicated an increase in caspase-8 activity.

Since the induction of necroptosis was also observed in SUIT-020 and Panc89 cells, we checked the assembly of the necrosome or complex IIb from these two cell lines upon CD95L/gemcitabine/zVAD-fmk, CD95L/gemcitabine/FLIPinB/zVAD-fmk, CD95L/gemcitabine/S63845/zVAD-fmk and CD95L/gemcitabine/FLIPinB/S63845/zVAD-fmk stimulations. Similar to the results obtained without zVAD-fmk, an increase in the amount of complex IIb was observed upon CD95L/gemcitabine/FLIPinB/S63845/zVAD-fmk stimulation compared to the other conditions in Panc89 cells as well as small increase in SUIT-020 cells (Fig. 9c,

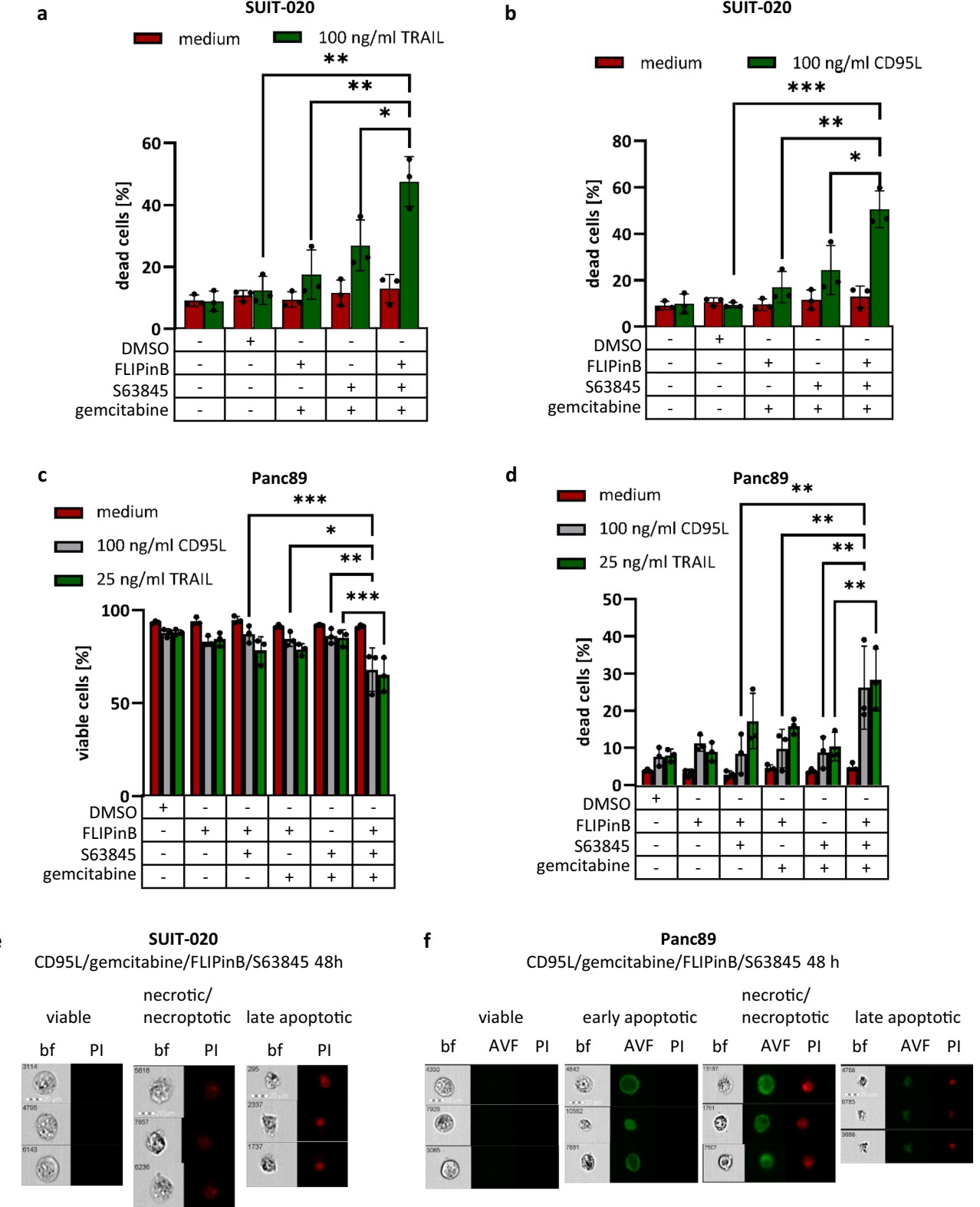

**Fig. 6 | DL/gemcitabine/FLIPinB/S63845 quadrupoule treatment shows the strongest cell death rate.** SUIT-020 cells (**a**, **b**) or Panc89 (**c**, **d**) were pretreated with 10 ng/ml gemcitabine for 24 h. Afterwards the cells were stimulated for 2 h with 20 µM FLIPinB and 10 (**a**, **b**) or 0.005 (**c**, **d**) µM S63845. Subsequently the cells were treated for 22 h with TRAIL (**a**, **c**, **d**) or CD95L (**b**–**d**). The cells were stained with PI only (**a**, **b**) or Annexin-V-FITC/PI (**c**, **d**). The amount of dead cells (**a**, **b**, **d**) or viable cells (**c**) was measured using Imaging Flow Cytometry. Mean and SD from three independent experiments are shown. The error bars indicate the mean ± SD. Corresponding values for viable cells of SUIT-020 are shown in Supplementary Fig. 5b, c, d. Representative images of SUIT-020 cells (**e**) or Panc89 (**f**) are shown. Cells were gated for PI negative (viable) and PI positive (late apoptotic and necroptotic) cells (**e**) or negative (viable), Annexin-V-FITC (early apoptotic), and Annexin-V-FITC/PI positive (late apoptotic and necroptotic) cells (**f**). For statistical analysis One-way ANOVA with Tukey post hoc tests were used to compare two conditions. The following values were used: ****$p < 0.0001$; ***$p < 0.001$; **$p < 0.01$; *$p < 0.05$; ns not significant. Abbreviations: bf bright field, AVF Annexin-V-FITC, PI Propidium Iodide.

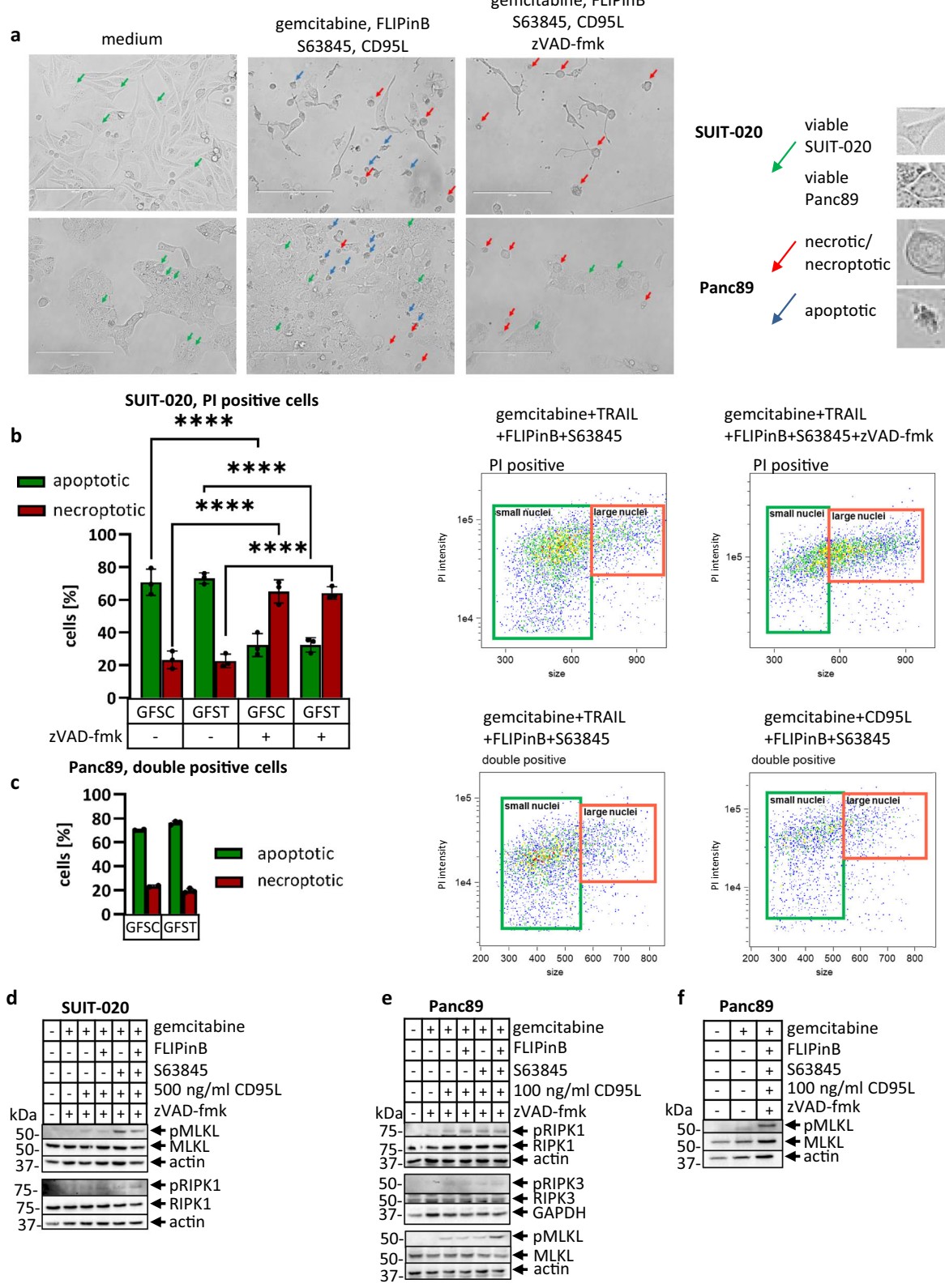

**a**

medium | gemcitabine, FLIPinB S63845, CD95L | gemcitabine, FLIPinB S63845, CD95L zVAD-fmk

SUIT-020 — viable SUIT-020 (green), viable Panc89 (green), necrotic/necroptotic (red), apoptotic (blue)

Panc89

**b** SUIT-020, PI positive cells ****

apoptotic / necroptotic

gemcitabine+TRAIL +FLIPinB+S63845 — PI positive

gemcitabine+TRAIL +FLIPinB+S63845+zVAD-fmk — PI positive

**c** Panc89, double positive cells

gemcitabine+TRAIL +FLIPinB+S63845 — double positive

gemcitabine+CD95L +FLIPinB+S63845 — double positive

**d** SUIT-020

**e** Panc89

**f** Panc89

Supplementary Fig. 10b–d). Specifically, an increase in FADD quantities as well as pRIPK1 in the complex IIb was detected upon quadrupuole treatment, which was in line with the increase of cell death upon administrations of all four stimuli. Taken together, it was shown that administration of DL together with gemcitabine, FLIPinB and S63845 leads to an increase in the assembly of complex II.

## Discussion

In this study, we analyzed the potential of targeting the key regulator of DL/gemcitabine network, the protein c-FLIP$_L$, in pancreatic cancer cells, which has been reported to be a promising target in this type of cancer. Since the combination of TRAIL with gemcitabine has been tested in clinical trials and presents a promising direction in the clinical research of PDAC[28], we tested

**Fig. 7 | DL/gemcitabine/FLIPinB/S63845 treatment induces necroptosis upon addition of zVAD-fmk to SUIT-020 and Panc89 cells. a** SUIT-020 cells or Panc89 cells were pretreated for 24 h with 10 ng/ml gemcitabine and afterwards for 2 h with 20 µM FLIPinB and 10 µM (SUIT-020) or 0.005 µM (Panc89) S63845 and for 1 h with 50 µM zVAD-fmk. Cells were treated afterwards for 22 h with 100 ng/ml CD95L. Representative microscopic images are presented with arrows showing cells with apoptotic morphology (blue arrows) and necroptotic morphology (red arrows). The quantification of apoptotic *versus* necroptotic cells in PI-positive populations for SUIT-020 (**b**) or Panc89 cells (**c**). The cells were pretreated for 24 h with 10 ng/ml gemcitabine and followed by 2 h stimulation with 20 µM FLIPinB and 10 (**b**) or 0.005 (**c**) µM S63845. Cells were afterwards treated for 22 h with 100 ng/ml CD95L or TRAIL. In addition, SUIT-020 cells were treated for 1 h before DL addition with 50 µM zVAD-fmk. The experimental data for this analysis is taken from Fig. 6 and

Supplementary Fig. 5. Representative gating strategy is shown on the right. Mean and SD from three independent experiments are shown. The error bars indicate the mean ± SD. SUIT-020 cells (**d**) or Panc89 cells (**e, f**) were pretreated for 24 h with 10 ng/ml gemcitabine and for 2 h with 20 µM FLIPinB and 10 (**d**) or 0.005 (**e, f**) µM S63845. Afterwards the cells were stimulated with indicated concentrations of CD95L for 5 h. In addition, cells were treated for 1 h before CD95L addition with 50 µM zVAD-fmk. Cell lysates were analyzed by Western Blot with the indicated antibodies. Actin served as loading control. One representative Western Blot out of three is shown. For statistical analysis One-way ANOVA with Tukey post hoc tests were used to compare two conditions. The following values were used: ****$p < 0.0001$; ***$p < 0.001$; **$p < 0.01$; *$p < 0.05$; ns not significant. Abbreviations: GFST gemcitabine/FLIPinB/S63845/TRAIL, GFSC gemcitabine/FLIPinB/S63845/CD95L.

the effects of targeting c-FLIP$_L$ together with co-administration of DL and gemcitabine. Furthermore, a compound targeting the key inhibitor of intrinsic cell death networks, Mcl-1 was also added to this study since the importance of simultaneous targeting c-FLIP and Mcl-1 has been suggested in the previous reports[22,29]. This quadrupole combinatorial treatment was shown to efficiently eliminate pancreatic cancer cells.

FLIPinB is the first-in class chemical compound targeting c-FLIP$_L$ which has been recently developed and tested in HeLa, Jurkat, and AML cell lines[12]. Previous studies have shown the major role for c-FLIP in PDAC suggesting testing the effects of targeting c-FLIP in this type of cancer[30]. The potential of FLIPinB in the elimination of pancreatic cancer cells in combinatorial treatments was explored in this study using three different pancreatic cancer cell lines. The efficiency of this compound strongly depends on the abundance of its target, c-FLIP$_L$, and c-FLIP$_L$ ratio to procaspase-8 in a particular cell line[12,35]. This, in turn, defines the amount of procaspase-8/c-FLIP$_L$ heterodimers that can be formed. The amount of heterodimers is also defined by the amount of the DISC or complex II formed in a particular cell line in response to the cell death stimuli[44]. In this study three cell lines had the different pattern of expression of the proteins of extrinsic apoptosis network with various expression levels of DRs, procaspase-8, and c-FLIP$_L$. Strikingly, FLIPinB had the highest activity on Panc89 cells, which was in accordance with the highest level of c-FLIP$_L$ expression in this cell line, which in turn supports our hypothesis on FLIPinB activity strongly depending on the levels of c-FLIP$_L$.

Among these three cell lines, the SUIT-020 cells had the highest level of XIAP expression. Accordingly, they were less sensitive to DL administration as well as to FLIPinB administration, supporting the critical role of the XIAP/caspase-3 ratio for the promotion of apoptosis initiated by caspase-8 activation[31,32,45,46]. Thus, the enhancement of caspase-8 activity at the DISC by FLIPinB in SUIT-020 cells was apparently not sufficient to induce apoptosis due to the blocking effects of XIAPs downstream of caspase-8 activation. This further supports the efficiency of our approach and requirement to target simultaneously different core components of cell death network like it was done in our study using the combination with targeting Mcl-1.

Importantly, in this study we have investigated three cell lines with different sensitivities towards DL and FLIPinB treatments and entirely different level of expression of core components of extrinsic cell death network. However, all three cell lines were sensitized towards cell death in the most efficient way by addition of all four compounds, which shows the potential efficiency of this combinatorial treatment. In this way, the combined targeting of c-FLIP$_L$ and Mcl-1 was shown to have a higher efficiency in enhancing DL/gemcitabine effects on inducing cell death. This is in line with other studies that show that simultaneously targeting Mcl-1 and c-FLIP can efficiently eliminate cancer cells[13,22].

Further, in our study we concentrated on the selection of low concentrations of stimuli that upon administration alone did not, or only marginally trigger cell death. Indeed, the goal was to apply low or threshold concentrations of stimuli which, when applied alone, do not induce cell death, but might trigger elimination of the cells when administered in combination. In this way, we aimed to prevent strong toxicity to normal, non-malignant cells, which is always a crucial aspect in cancer therapy. We have shown that a quadrupole combination of the stimuli applied in the low concentrations was

efficient and resulted in the elimination of pancreatic cancer cells, which can be considered in the future research for developing anti-cancer therapies.

c-FLIP and Mcl-1 are characterized by a short half-life. Hence, there is an increasing attention in the anti-cancer therapies towards compounds that are directed towards blocking the expression of these two proteins and thereby causing apoptosis induction[20,47]. In regard of c-FLIP and Mcl-1 downregulation, the promising directions involve the application of microRNAs (miRNAs), histone deacetylase (HDAC) inhibitors, inhibitors of cyclin-dependent kinase 9 (CDK9) and translational inhibitors including the ones obtained from natural products[47,48]. Further, several reports have shown that treatment with gemcitabine in high concentrations results in the downregulation of c-FLIP proteins, thereby promoting cell death[29]. Accordingly, in this study, we selected the lower concentration of gemcitabine than the one that was reported to cause a strong downmodulation of c-FLIP levels[29]. The question whether a direct targeting of c-FLIP and Mcl-1 with chemical compounds is more efficient for the particular anti-cancer treatment compared to downregulation of c-FLIP and Mcl-1 levels has to be addressed in the future studies.

The unexpected observation made in this study is the strong increase in the formation of the complex II upon DL/gemcitabine treatment, which is observed upon administration of both inhibitors, FLIPinB and S63845 as well as FLIPinB alone. As mentioned above, changes in the expression of any of the core components of this complex were not detected upon FLIPinB, S63845 or their combined action. Moreover, gemcitabine has been reported to cause an increase in RIPK1 and RIPK3 amounts in pancreatic cancer cells, which was also not observed in our study[27]. Hence, molecular mechanisms of the increase of the assembly of complex II remain open. It might be suggested that the enhancement of caspase-8 activity at the DISC by FLIPinB leads to a faster complex II formation and that the enzymatic activity of caspase-8 promotes complex assembly. This would be supported by the increase of caspase-8 activity due to the caspase-3-mediated feedback loop mediated through inhibitory effects on Mcl-1 (Fig. 10). The other possibility involves the stabilization of the complex II by FLIPinB. The latter might be mediated *via* FLIPinB-mediated stabilization of the procaspase-8/c-FLIP$_L$ heterodimer, which has been suggested to serve as a scaffold for the complex II formation recently[49]. This also asks for implementation of mathematical modeling to better understand the quantitative regulation of this pathway[44].

Taken together, our studies have shown the possibility of simultaneously targeting two key players of the apoptotic network in pancreatic cancer: c-FLIP$_L$ and Mcl-1 upon DL/gemcitabine treatment. In this regard, the activity of FLIPinB compound targeting c-FLIP$_L$ has been investigated for the first time on PDAC. Further, we delineated several features of this combinational treatment leading to the enhancement of cell death including the increase in complex II formation. This strategy paves the way towards more targeted therapeutic approaches in cancer.

## Methods
### Cell culture
Pancreatic cancer SUIT-020 cells were maintained in DMEM/Hams-F12 media (PAN Biotech, Germany). MiaPaca2 cells, Panc89 cells and HT29 cells were maintained in DMEM media (Thermo Fisher Scientific Inc.,

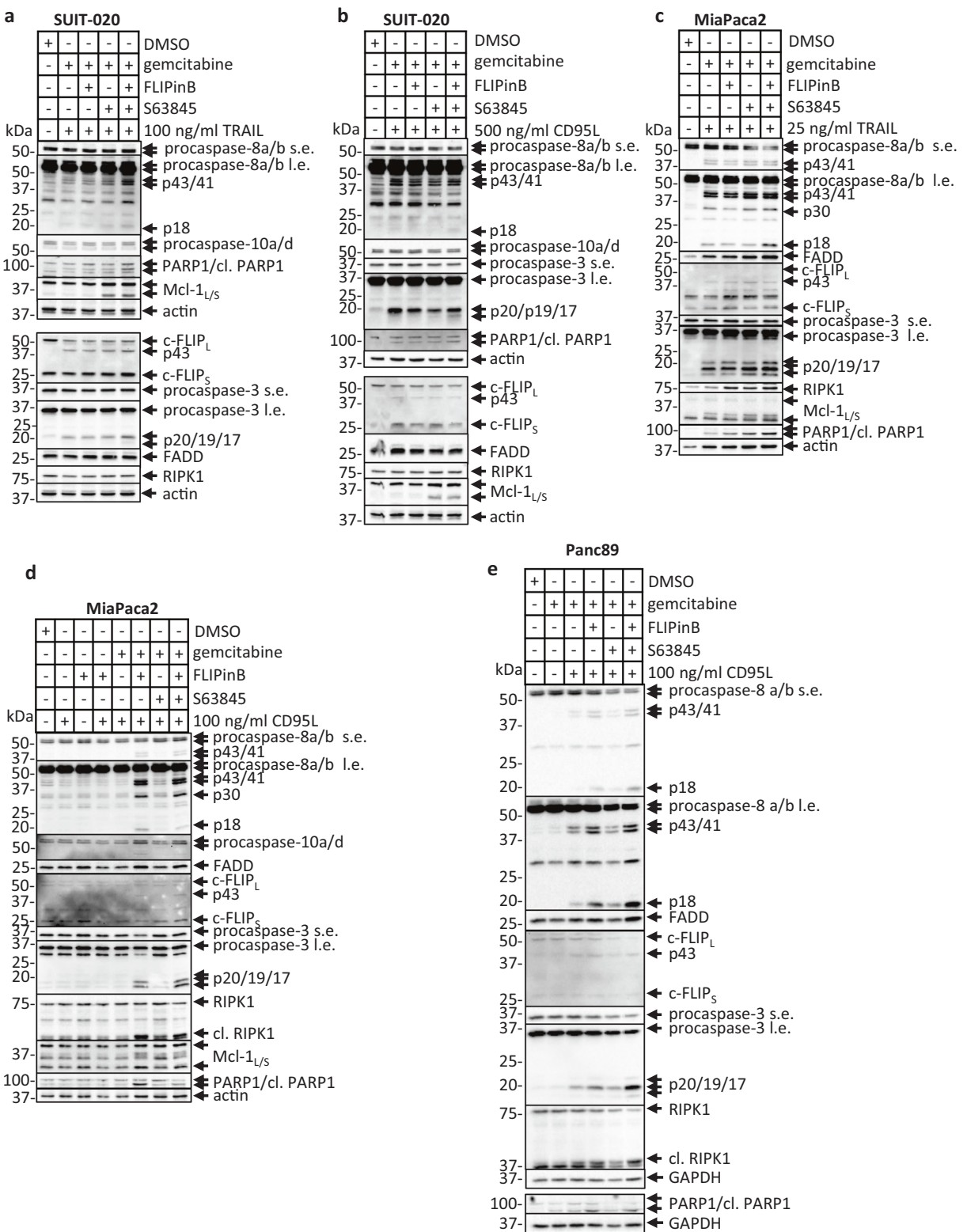

**Fig. 8 | DL/gemcitabine/FLIPinB/S63845 treatment enhances the cleavage of caspases and their substrates.** SUIT-020 cells (**a**, **b**), MiaPaca2 cells (**c**, **d**) or Panc89 cells (**e**) were pretreated for 24 h with 10 ng/ml gemcitabine and for 2 h with 20 (**a**, **b**, **e**) or 50 (**c**, **d**) μM FLIPinB and 10 (**a**, **b**), 2.5 (**c**, **d**) or 0.005 (**e**) μM S63845. Afterwards the cells were stimulated with indicated concentrations of TRAIL (**a**, **c**) for 3 h (**a**, **c**) or 6 h (**a**) or CD95L (**b**, **d**, **e**) for 3 h (**e**), 5 h (**d**) or 6 h (**b**). Cell lysates were analyzed by Western Blot with the indicated antibodies. Actin or GAPDH served as loading control. One representative Western Blot out of three is shown. Abbreviations: s.e. short exposure, l.e. long exposure.

USA). Human fibroblasts were maintained in RPMI 1640 (Thermo Fisher Scientific Inc., USA). 10% heat inactivated fetal calf serum and 1% Penicillin/Streptomycin was added to the media. Cells were cultured in 5% $CO_2$ and at 37 °C. SUIT-020 and MiaPaca2 cells were the kind gift of Prof. N. Giese

(University of Heidelberg); HT29 cells were the kind gift of Prof. Thomas Brunner (University of Konstanz). Panc89 cells and primary fibroblasts were from National Center for Tumor Diseases (TU-Dresden). The testing for mycoplasma were performed once in 4 weeks.

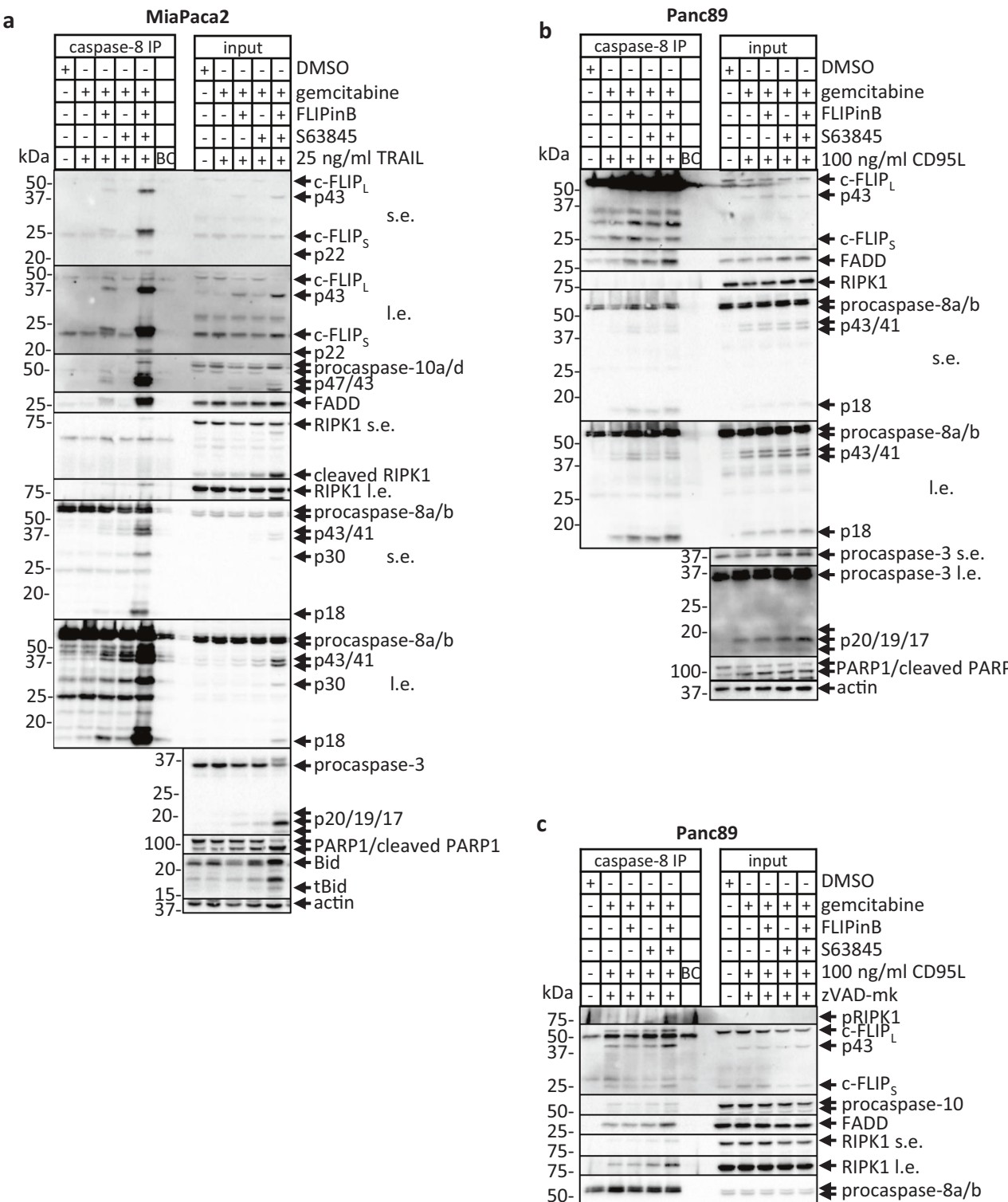

**Fig. 9 | DL/gemcitabine/FLIPinB/S63845 treatment enhances complex II assembly.** MiaPaca2 (**a**) or Panc89 (**b**) cells were pretreated with 10 ng/ml gemcitabine for 24 h and subsequently with 2.5 (**a**) or 0.005 (**b**) μM S63845 and 50 (**a**) or 20 (**b**) μM FLIPinB for 2 h. Afterwards the cells were treated with TRAIL (**a**) or CD95L (**b**). **c** Panc89 cells were pretreated with gemcitabine for 24 h and subsequently with 0.005 μM S63845 and 20 μM FLIPinB for 2 h. Afterwards the cells were treated with CD95L for 5 h. 1 h prior CD95L treatment cells were treated with 50 μM zVAD-fmk. The immunoprecipitation was carried out with anti-caspase-8 antibodies (caspase-8 IP). The caspase-8 IP was analyzed using the corresponding antibodies. Western Blot analysis of caspase-8 serves as a loading control for caspase-8 IP. Total cell lysates were analyzed, additionally (input). Actin served as loading control for total cell lysates. One representative Western Blot out of three is shown. Abbreviations: s.e. short exposure, l.e. long exposure, IP immunoprecipitation, BC Bead control pulldown.

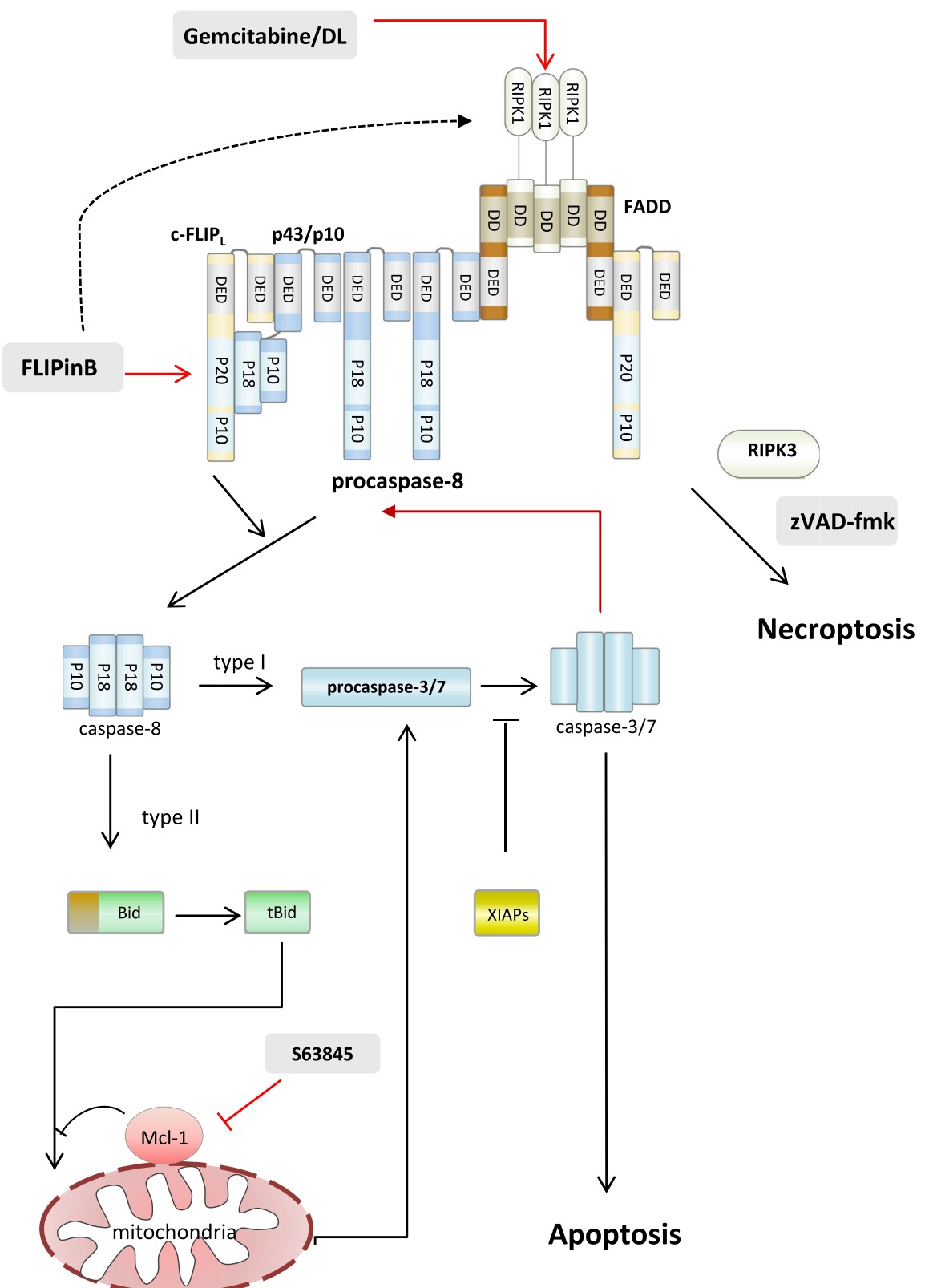

**Fig. 10 | Scheme of FLIPinB effects on the DL/gemcitabine network.** The formation of the complex II upon DL/gemcitabine treatment is shown leading to caspase-8 activation. Caspase-8 activates caspase-3 *via* type I or type II signaling pathways, which might be inhibited by XIAPs. The putative stabilization effects of FLIPinB on complex II are shown with an arrow.

## Antibodies and reagents

The following antibodies were used for Western Blot analysis: polyclonal anti-actin antibody (A2103) from Sigma-Aldrich, Germany; monoclonal anti-Bak antibody (#12105), monoclonal anti-Bax antibody (#5023), monoclonal anti-Bcl-10 (#4237) polyclonal anti-Bid antibody (#2002), polyclonal anti-caspase-3 antibody (#9662), monoclonal anti-caspase-9 (#9508), monoclonal anti-cytochrome c (#11940) monoclonal anti-DR4 antibody (#42533), monoclonal anti-DR5 antibody (#8074), monoclonal

anti-LC3B antibody (#3868), monoclonal anti-RIPK1 XP antibody (#3493), polyclonal anti-PARP antibody (#9542), monoclonal phosphor-RIPK3 (Ser227) (#93654), monoclonal MLKL (#14933), monoclonal anti-XIAP (#14334) from Cell Signaling Technology, USA; monoclonal anti-Bcl-2 antibody (sc-7382), polyclonal anti-CD95 antibody (sc-715), monoclonal anti-GAPDH (sc-47724), polyclonal anti-Mcl-1 antibody (sc-819) from Santa Cruz, USA; monoclonal anti-Bcl-x antibody (610209) from BD Transduction Laboratories, USA; monoclonal anti-caspase-10 antibody (M059-3) from MBL International Corporation, USA; monoclonal anti-pMLKL (phospho S358) (ab187091) and polyclonal anti-RIPK3 (ab226297) from abcam, UK. Horseradish peroxidase-conjugated goat anti-mouse IgG1,-2b, and goat anti-rabbit were from Southern Biotech, USA. Recombinant TRAIL (KillerTRAIL™) was from Enzo Life Sciences, Germany. All chemicals were of analytical grade and purchased from Merck (Germany) or Sigma (Germany). ABT-263 (A10022-10) was from Hölzel Diagnostika (Germany). The S63845 was from APExBIO (A8737, USA), FLIPinB was from Ambinter (Amb1202053, France), zVAD-fmk (BAC-4026865.0005) was from Biozol (Germany); Nec-1s (5042970001) and GSK872 (5.30389) were from Merck (Germany). Gemcitabine-hydrochloride was from Merck (G6423, Germany). The anti-APO-1, monoclonal anti-FADD (clone 1C4), monoclonal anti-caspase-8 (clone C15) and monoclonal c-FLIP (clone NF6) antibodies were kindly provided by Prof. P. H. Krammer (DKFZ, Heidelberg). Recombinant LZ-CD95L was produced as described[50,51]. For receptor surface staining FITC anti-human CD95 (Fas) (#305605), PE anti-human CD261 (DR4, TRAIL-R1) (#307205), PE anti-human CD262 (DR5, TRAIL-R2) (#307405) antibodies from BioLegend (USA) were used.

### Treatment of the cells with CD95L/TRAIL and/or inhibitors

SUIT-020, MiaPaca2 and Panc89 cells were always seeded 1 day before experiment. Before stimulation, fresh medium was given to the cells, while the old one was discarded. The cells were treated for 24 h with gemcitabine and subsequently for 2 h with FLIPinB and/or S63845/ABT-263 in the indicated concentrations. Two different batches of S63845 from APExBIO (A8737, USA) were used: the batch 6 for the experiments with SUIT-020 and MiaPaca2 cells, and the batch 9 for the experiments with Panc89 cells. Afterwards, CD95L or TRAIL was added for indicated time intervals. The cells were not washed in-between stimulation steps and medium was not changed in the subsequent steps.

### Cell viability assay

$1.2 \times 10^4$ SUIT-020, MiaPaca2 or Panc89 cells per well were seeded the day before stimulation in 96 well plates. Measurement of ATP content was in accordance to manufacturer's instructions (CellTiter-Glo® Luminescent Cell Viability Assay, Promega, Germany). The luminescence intensity was measured by the microplate reader Infinite M200pro (Tecan, Switzerland) in duplicates. Values from treated cells were normalized to the values of untreated cells, e.g., "medium control". Medium control was set as 100% cell viability.

### Caspase-3/7 activity assay

$1.2 \times 10^4$ SUIT-020, MiaPaca2 or Panc89 cells per well were seeded the day before stimulation in 96 well plates. Measurement was in accordance to manufacturer's instructions (Caspase-Glo® 3/7 Assay, Promega, Germany). The luminescence intensity was measured by the microplate reader Infinite M200pro (Tecan, Switzerland) in duplicates. Values from treated cells were normalized to the values of untreated cells, e.g., "medium control". Medium control was set as one RU (relative unit) as described previously[12,52].

### LDH assay

$1.2 \times 10^4$ SUIT-020, MiaPaca2 or Panc89 cells per well were seeded the day before stimulation in 96 well plates. Measurements were carried out in accordance to manufacturer's instructions (LDH-Glo™ Cytotoxicity Assay, Promega, Germany). The luminescence intensity was measured by the microplate reader Infinite M200pro (Tecan, Switzerland) in duplicates.

Values were normalized to medium control. Medium control was set as one RU (relative unit).

### 3D-culture

$6 \times 10^3$ Panc89 cells were seeded in Biofloat™ plates (Sarstedt, Germany) in 50 µl medium. After 4 days, medium was removed, and cells were treated as described previously. The formation of spheroids was validated via microscopy. The microscope was an EVOS FL imaging system (Thermo Fisher Scientific Inc., USA) and the $20 \times$ magnification was used to take pictures.

### Western Blot analysis of total cell lysates

The Western Blot analysis of total cell lysates was performed in accordance with our previous reports[13,53]. Analysis was performed using Image Lab™ 5.1 Software (BioRad).

### Caspase-8-immunoprecipitation (C8-IP), DISC-immunoprecipitation (DISC-IP) and FADD-immunoprecipitation (FADD-IP)

$5 \times 10^6$ SUIT-020, $3 \times 10^6$ MiaPaca2, or $8 \times 10^6$ Panc89 cells were seeded the day before stimulation in 14.5 cm plates. The cells were harvested as described before[52]. Before Immunoprecipitation, one/tenth of the total cellular lysates was taken as lysate/input control. 10 µl of Protein-A Sepharose beads and 2 µg of anti-Caspase-8 antibodies (clone C15), anti-FADD (clone 1C4), or anti-APO-1 antibodies were added to the total cellular lysates and incubated gently shaking over night at 4 °C. For the control of unspecific binding, the total cellular lysates were incubated with Protein-A Sepharose beads only (bead control pulldown (BC)). Protein-A Sepharose beads were washed four times with Phosphate buffered saline (PBS).

### TNF-α ELISA

$1.2 \times 10^4$ SUIT-020 cells per well were seeded the day before stimulation in 96 well plates. Human TNF-α was measured using Human TNF-α ELISA MAX™ Standard Set (BioLegend, USA). Preparation of ELISA plate was in accordance to manufacturer's instructions. Absorbance was measured using the microplate reader Infinite M200pro (Tecan, Switzerland) in duplicates. With every ELISA a standard curve was generated using 500 pg/ml TNF-α stock standard solution and six two-fold serial dilutions were performed. The standard curve was plotted with a fifth degree polynomial. Results from measurement were estimated using the standard curve.

### Imaging flow cytometry

$1 \times 10^6$ SUIT-020 cells were stained with PE anti-human TRAIL-R1 or 2 antibodies or FITC anti-human CD95 antibodies (BioLegend, USA) according to manufacturer's instructions. Stained and unstained cells were analyzed using Imaging Flow Cytometry (AMNIS®).

For cell death measurements, $0.2 \times 10^6$ SUIT-020 cells or $0.5 \times 10^6$ Panc89 cells were seeded the day before stimulation in 6 well plates. Cells were stained with Annexin-V-FITC, PI, or their combination in accordance with our previous studies[42,54]. For estimation of apoptotic versus necroptotic cells the gating was carried out based on the size of the PI-stained nucleus as described previously[42].

### Gel filtration followed by DISC IP

$8 \times 10^6$ SUIT-020 cells were seeded the day before stimulation in 14.5 cm plates. Three 14.5 cm plates were used per one condition. Cells were harvested and lysed as described above. Äkta™ pure (GE Healthcare, Germany) was used. Column Superose™ 5 10/300 GL (GE Healthcare, Germany) was prepared and filled with the lysis buffer. With a syringe 500 µl lysate was given in a 250 µl loop on the column and fractionated at 4 °C with a pump speed of 0.2 ml/min. Fraction 1 was discarded. The other fractions were merged in the following way: 2–6, 7–10, 11–15, 16–20 and 21–25. 10 µl Protein-A Sepharose beads and 2 µg of anti-APO-1 antibodies were added to the fractions and incubated over night at 4 °C with rotation. Protein-A Sepharose beads were washed four times with Phosphate buffered saline (PBS).

## Statistics

For statistical analysis unpaired One-way ANOVA with Tukey post hoc tests were used to compare two different conditions, which is shown in diagrams with a bracket. One-way ANOVA tests were used to compare a group of conditions. This is shown in diagrams with a line over all the conditions which were involved in the statistical analysis. The statistical analysis was calculated with GraphPad prism 8 software. The following values were used: $****p < 0.0001$; $***p < 0.001$; $**p < 0.01$; $*p < 0.05$; ns not significant.

## Reporting summary

Further information on research design is available in the Nature Portfolio Reporting Summary linked to this article.

## Data availability

All data are available from the corresponding author on reasonable request. Numerical source data for all graphs in the manuscript can be found in Supplementary Data file 1.

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

## Acknowledgements
We acknowledge the Wilhelm Sander-Foundation (2017.008.02), Budget project FWNR-2022-0020 on systems biology and bioinformatics, project ALBB (European Regional Development Fund) and the DFG (LA 2386) for supporting our work. We thank Dr. Anna Dittrich, OvGU, for help with experiments. Corinna König was supported by fellowship of OvGU.

## Author contributions
C.K. performed experiments, analyzed the data, and contributed to the manuscript text; N.I. performed data analysis; V.I. and D.K. contributed to technologies and contributed to the manuscript text; I.L. supervised the study and wrote the manuscript.

## Funding

## Competing interests
The authors declare no competing interests.
