## [Transparent Peer Review file · Communications Biology]

Pharmacological targeting of caspase-8/c-FLIPL heterodimer enhances complex II assembly and elimination of pancreatic cancer cells

Corresponding Author: Professor Inna Lavrik

This manuscript has been previously reviewed at another journal. This document only contains information relating to versions considered at Communications Biology.

Version 0:

Reviewer comments:

Reviewer #1

(Remarks to the Author)

In the manuscript "Pharmacological targeting of c-FLIPL enhances cell death via increase of complex II assembly in pancreatic cancer cells", Corinna Konig et al., demonstrate synergy in targeting Death receptor pathways in combination with gemcitabine and Mcl-1 inhibitor.

I think the data is interesting and worth considering but need further improvements. I am particularly concerned and not convinced about the conclusions on the necroptosis driven by combining Trail/CD95 with Mcl-1 inh and gemcitabine. Additionally, I would like to see more relevant 3D model system of pancreatic cancer/normal included in the study.

Major points:

1. Conclusions in the paper are based on two pancreatic cancer cell lines and it would be good if the authors included a panel of pancreatic cell lines to see how general these findings are. One of the cell lines SUIT-020 are resistant to death ligands, could that be explained? Is it something to do with death machinery (apoptotic/necroptotic) or it is result of the death receptor activation? This can be addressed by using positive control: treating the cells with TNF and Smac mimetic (for apoptosis) or TNF, Smac mimetic and zVADfmk (for necroptosis).
2. Authors are stating: "In our study, we have aimed at selecting the lowest concentrations of the cell death inducing stimuli which when applied alone do not induce any cytotoxic effects but work in combination to eliminate cancer cells. In this way, we aimed to mimic the effects for reducing the possible side effects of combinatorial treatment using cell lines as a model system." Definitely there is a logic to minimise toxicity by reducing the dose and combining with drugs that target different pathways, however there is no evidence in the work presented about the response of normal, non tumour cells. How selective this combination is for pancreatic cancer cells? One way would be to expand the study using pancreatic derived cancer organoids and compare them with normal organoids. Such model system will increase the strength of this study.
3. In figure 1 a-it is good to indicate each line what represent. It is described in the text however the figure should be clear. Additionally RIPK3 levels are not that conclusive. It is very important to draw conclusions since necroptosis is studied. Authors need to use a positive control for RIPK3 expression so we can have an idea what is the level of RIPK3. Such positive control is HT29 that has been studied extensively in the necroptosis field and I would like to see what is the relative level of RIPK3 expression between these cell lines. And as I suggested above, positive control of necroptosis using TNF+Smac mimetic+zVADfmk for SUIT-020 and if HT29 could be included as control. It will be important so we know how necroptosis works in SUIT-020.
4. In figure 1 b-e, I would suggest siRNA against FADD be used to demonstrate specificity.
5. In suppl fig 2a-d-why there is no more cleaved casp8/casp3 when combination of gemcitabine and TRAIL is used? What is the evidence that casp8 activation drives this death?
6. Fig 3d-e- The effect of triple combination is not very convincing because double (DL/gemcitabine) can cause similar effect.
7. TNF is not classical NFkB gene and is more regulated by p38. I would suggest the authors to use some of the following

NFkB genes: IL6/SOD2/BIRC3/A20.

8. Fig 5e- This is not accurate assay to distinguish between apoptosis and necroptosis. The issue is that there are many cells that are not 100% apoptotic or necroptotic and that will make very difficult to split the percentage of apoptosis versus necroptosis. The way to demonstrate whether the cells can die by necroptosis is by using different inhibitors like RIPK1 inh and caspase inh. And in figure 6 authors did use them. However I am not convinced that there is necroptosis since zVADfmk in presence of Trail/CD95 protects. If that was the case, it should sensitise to cell death-necroptosis. Additionally, I would suggest the authors to use more selective inhibitors of RIPK1 like GSK'963 or PK68 that work in nM concentrations rather than Nec1S that is not very selective. Apart of cell death assay it will be important to show necrosome formation by immunoprecipitation and activation of RIPK3/MLKL on western blot.

Reviewer #2

(Remarks to the Author)

Cancer cells are typically characterized by their ability to resist programmed cell death (PCD) pathways, promoting tumor survival. This is often achieved through loss or decreased expression of PCD signaling pathway components, such as enzymes required for classical apoptosis or alternative PCD forms such as necroptosis. Strategies to restore or enhance PCD pathways in tumor cells, including the design of novel therapeutics with fully characterized mechanisms of action, represent an important area of cancer research with clear translational relevance. This group had previously reported the design of a novel "FLIP interactor" molecule (FLIPinB) that functioned by binding to and stabilizing c-FLIPL, an important pro-apoptotic regulator of extrinsic apoptosis. Because c-FLIPL is a component of the death-inducing signaling complex (DISC) that binds to pro-caspase-8, this stabilizing interaction between the c-FLIPL and caspase-8 heterodimer promotes the cleavage of pro-caspase-8 into active caspase-8, resulting in extrinsic apoptosis induction following activation of death receptors such as CD95 or TRAIL. Despite these intriguing findings, the complete molecular mechanisms of FLIPinB activity, particularly in relevant cancer cell lines rather than HeLa cells, remained incompletely understood. Establishing such information about FLIPinB action would provide essential information needed to realistically assess the value in carrying this drug candidate forward as a novel therapy in future translational studies.

Here, Konig and colleagues expand upon their previous work by providing a detailed overview of FLIPinB's molecular mechanism of action. Using 2 pancreatic ductal adenocarcinoma (PDAC) cell lines with differing death receptor (DR) ligand sensitivities, they show that combinatorial treatment of various death modulators with FLIPinB increases pancreatic tumor cell death via apoptotic and necroptotic signaling pathways, and that this is due to an increase in pro-death complex II assembly and activation.

This study is thorough and well-written, and addresses an interesting area of cancer research with potential to inform novel translational studies for pancreatic cancer. There are a few aspects of the study that I believe would benefit from further experimental investigation and/or discussion in order to strengthen the authors' conclusions and overall impact of the manuscript. Here is a summary of my primary comments:

1. Showing specificity of death effects in pancreatic tumor cells vs. healthy cells

o The majority of experiments are performed in 2 PDAC cell lines in culture: MiaPaca2 as a model sensitive to extrinsic apoptosis via DR ligands, and SUI-020 as a DR ligand-resistant model. However, it is unclear whether the combination therapies tested also have cytotoxic effects on non-cancerous cells, including healthy pancreatic cells.

o The authors shouldn't be asked to repeat all of these experiments in non-cancerous cells, as this represents a considerable amount of extra work. However, it would be helpful to see cell viability data for either (a) a healthy pancreatic cell line (such as HPDE6c7, Millipore #SCC442) or (b) primary human pancreatic epithelial cells (such as Accegen #ABC-TC3744) treated with gemcitabine + death receptor ligand (CD95L or TRAIL), +/- FLIPinB. If these combinatorial treatments specifically lead to tumor cell death (as seen in MiaPaca2 cells in Fig. 2E) with minimal observed cytotoxicity in healthy cells, this would considerably strengthen the translational value of the manuscript by allowing claims regarding the potential safety of combination therapies in future translational studies. If the combination therapies induce equivalent amounts of cell death in non-transformed cells, this should be included as a potential source of off-tumor toxicity in translational studies.

2. Points related to programmed cell death pathways in PDAC cell lines treated with combination therapy

o To characterize the mechanism of programmed cell death (PCD) by which SUI-020 cells are dying following combination treatment, the authors first show morphological changes + PI uptake via AMNIS. The sample images shown in Fig. 5E look convincing with respect to necroptotic and late apoptotic cell identification, but it would be helpful to provide quantification of what percentage of cells analyzed fall into each category (viable vs. necroptotic vs. late apoptotic) to add information about cell death dynamics on a population level. It would also be helpful to include more information about the exact treatment conditions of these cells and the timepoint at which they were harvested for flow cytometry; do these correspond to Fig. 5C (gemcitabine + TRAIL + FLIPinB + S63845) or Fig. 5D (gemcitabine + CD95L + FLIPinB + S63845)?

o The results section for Figure 6A/B states that SUI-020 cells die through a combination of necroptosis and apoptosis, using a set of experiments using stimulation with either TRAIL (Fig. 6A) or CD95L (Fig. 6B) with combination therapy +/- the RIPK1 inhibitor necrostatin-1s (Nec-1) or the pan-caspase inhibitor zVAD-fmk. Although the authors state that Nec-1s or zVAD-fmk administration alone leads to a "partial rescue" of SUI-020 viability, these differences (white bars in 6A and 6B, 3rd and 4th sets of bar graphs) are not statistically significant and should be discussed as equivalent amounts of cell viability. Only combination treatment with Nec-1s + zVAD-fmk significantly restores cell viability to that observed with DMSO vehicle treatment (white bars in the 1st set of bar graphs).

o Nec-1s is used in Fig. 6, but this drug inhibits RIPK1; while RIPK1 activation is involved in necroptosis, it can also serve

death-independent signaling functions if RIPK3 and MLKL are not activated downstream of RIPK1 (reviewed by Newton K, Cold Spring Harb Perspect Biol 2020). Claims regarding necroptosis would therefore be strengthened by revision experiments providing additional molecular characterization of the activation of different necroptosis signaling components. This would be particularly helpful for SUIT-020 cells, which express both caspase-8 and RIPK3 (Fig. 1A) and should be capable of undergoing both forms of PCD. Additional information could include Western blots for phospho-RIPK3 and/or phospho-MLKL (only found in necroptotic cells) in wild-type SUIT-020 cells, or genetic tools to knock out or knock down necroptotic signaling machinery in SUIT-020 cells and seeing how their deficiency influences cell viability. These types of tools could be used to answer the following questions:

- a) Does combination therapy (gemcitabine + TRAIL/CD95L + FLIPinB + S63845) lead to an increase in phospho-RIPK3/phospho-MLKL in wild-type cells? Is the presence of these phosphorylated necroptosis pathway components blocked following administration of Nec-1s?
- b) Does removal of necroptotic signaling components influence SUIT-020 viability following combination therapy administration (gemcitabine + TRAIL/CD95L + FLIPinB + S63845)? Such manipulations could include generating RIPK3^{-/-} or MLKL^{-/-} SUIT-020 cells using CRISPR/Cas9, or knocking down gene expression using stable transduction of shRNA/transient transfection with siRNA.

3. Modeling caspase-8:c-FLIP ratio in PDAC samples beyond MiaPaca2 and SUIT-020 lines

o The authors cite a previously published mathematical model (Ivanisenko & Lavrik, Biochemistry 2020) that they use to predict the potentially efficacy of FLIPinB treatment, based on the ratio of c-FLIPL to pro-caspase-8 expression in target cells. While they show where MiaPaca2 and SUIT-020 cell lines fall according to their model (Fig. 2B), it is unknown how these results compare to more clinically relevant PDAC patient samples. Since FLIPinB efficacy requires an optimal ratio of 2:1-3:1 pro-caspase-8:c-FLIPL and therefore may not work in a context of high pro-caspase-8/low c-FLIPL content (i.e. settings where there are lower amounts of pro-caspase-8/c-FLIPL heterodimers for FLIPinB to stabilize), it could be helpful to provide data showing that the ratios observed in MiaPaca2 and SUIT-020 cell lines fall within a similar range to that observed in primary patient samples.

o In the absence of access to primary human PDAC samples, if this group's model can use proteomics data as an input, they may consider analyzing published proteomics data (such as Cao et al, Cell 2021) to see where primary human PDAC cells (and healthy adjacent tissues) fall according to their model.

o This represents a fair amount of computational work and I do not know if the group's model is even compatible with the data outputs available from published proteomics data sets. However, since the likely efficacy of FLIPinB depends on the ratio of pro-caspase-8 and c-FLIP found in tumor cells, if it is possible to characterize human PDAC samples using this model then this may provide helpful data lending strong justification for pursuing FLIPinB combinatorial therapy in future translational studies.

o In the absence of any additional computational work, the authors may consider including a brief discussion of the work presented in Schmid SJ et al. (Pancreas 2013) showing that a majority of PDAC surgical resections were c-FLIP positive. Although this does not incorporate caspase-8 expression data, the fact that the majority of PDAC patients likely maintain intact c-FLIP expression could be promising when considering potential patient responsiveness to FLIPinB combination therapy.

In addition to the primary comments listed above, I also have a handful of minor criticisms that I believe would strengthen the manuscript if addressed by the authors:

1. Model figure readability

o The model figure presented in Fig. 8B is comprehensive but very busy. The authors may consider revising the model to remove several signaling components that are not directly examined in this manuscript (i.e. remove BID-tBID, XIAP, caspase-9, SMAC/DIABLO, MOMP, cytochrome-c), which may help simplify and highlight the key findings of the paper.

2. Polishing figure labels

o There are a few instances where figures could be updated be labeled more consistently. These include:

a) Fig. 3A, 3B: The differing font sizes in the table row for S63845 are distracting; all font sizes could be updated to be the same smaller size.

b) Fig. 3C: The concentrations for S63845 are written using “,” comma notation rather than using “.” decimal notation. To be more consistent with concentrations listed in other paper figures, please update to decimal notation.

Reviewer #3

(Remarks to the Author)

Corinna König et al. described here that combined treatment with DL, gemcitabine, MCL-1 inhibitor S63845 and FLIPinB could efficiently decrease pancreatic cancer cell viability with relatively low dosages by inducing both apoptosis and necroptosis. While this is potentially interesting as a therapeutic strategy to treat pancreatic cancer patients, there are several main issues to be further clarified or evaluated. Moreover, the mechanism of this treatment-induced cell death needs to be further investigated. I therefore have the following comments that need to be addressed.

Major comments:

1. This study only tested the treatment in two representative pancreatic cancer cell lines MiaPaca2 and SUIT-020 cells. Since all the drugs used here are somewhat toxic to the cells, the authors need to show whether this treatment also affect the healthy normal cells in addition to cancer cells.

2. I am not sure whether it is realistic to apply four toxic drugs in vivo to treat cancer patients. Particularly, it is important to minimize the side effects of these drugs, therefore, even if this is possible, the concentrations of the drugs need to be carefully tested and optimized again in vivo. Therefore, I would believe that performing in vivo experiments to prove that the treatment works in vivo would be necessary.

3. The FLIPinB affects extrinsic apoptosis pathway, while the MCL-1 inhibitor affects intrinsic apoptosis pathway, combined treatment of these two drugs with DL/gemcitabine therefore would affect both intrinsic apoptosis and extrinsic apoptosis. Thus, the authors should examine Caspase-9 cleavage as well. To more precisely address the mechanism, the authors should perform the experiments in Caspase-8 and Caspase-9 KO cells. Moreover, the authors only showed the happening of necroptosis by inhibiting RIPK1 kinase activity without showing any necroptosis markers activation, such as p-RIPK1, p-RIPK3 and p-MLKL in the lysates. The author should at least examine the necroptosis markers activation. Plus, I am not sure whether the very low (3-8pg/mL) TNF production presented in Figure 4d is meaningful or not.

4 The effect of majority of the data are marginal and not obvious in my opinion. For example, in Figure 7a, CI-Caspase-3 and CI-Caspase-8 in TGSF-treated samples are not obviously higher than TSF-treated samples; in Figure 7b, CI-Caspase-3 in CGSF-treated sample is even lower than CG-treated sample; Honestly, I could not tell the difference between necroptosis and late apoptosis from the pictures presented in Figure 5e; in Supp-Figure 2, the difference of CI-Caspase-3 and CI-Caspase-8 are not obvious, even though the p17/19 of CI-Caspase-3 are increased in TGSF- or CGSF-treated samples, gemcitabine treatment alone already induced similar amount of CI-Caspase-3 activation; in Supp-Figure 7a, the DISC components recruited to the complex are less in CGSF-treated samples than CGF-treated samples and similar with CSF-treated samples, which suggests that FLIPinB does not further increased DISC formation; the effect in Supp-Figure 8b is marginal.

Minor comment:

1 Which groups are compared for statistical analysis in Figure 1? Control groups with all the other groups by paired-comparison with the same p-value or one p-value by one-way ANOVA is labelled on the graph?

2 The variation in some experiments are too big to draw a precise conclusion. For example, in Figure 3a, the experiment need to be redo to draw a conclusion that S63854 does not induce cell death in SUIT-020 cells.

3 In Figure 3C, the authors should compare CI-Caspase-3 activation between MiaPaca2 and SUIT-020 cells and include a positive control (CI-Caspase-3 positive lysates) to make a conclusion that the low sensitivity of SUIT-020 cells towards S63845 treatment was also observed in Western Blot analysis of caspase cleavage.

Version 1:

Reviewer comments:

Reviewer #1

(Remarks to the Author)

Dear Isobel,

I checked the response of the authors to my comments and I am happy that they included my suggestions and addressed my concerns. So I am happy for the paper to be accepted.

Best wishes

Reviewer #2

(Remarks to the Author)

Overall I am impressed by the amount of work put into the revision of this manuscript and I feel that the authors put strong effort into addressing each of my initial critiques/suggestions. I have listed any comments to individual points below:

(1) Specificity of death effects in PDAC cells versus healthy cells. The authors repeated numerous assays in human fibroblasts and show minimal toxicity in non-cancerous cells. They have also included a third PDAC line (Panc89) in their panel of key experiments which helps further strengthen their conclusions.

(2) Further elucidation of PCD pathways engaged in PDAC cell lines.

(2a) The quantification of apoptotic versus necroptotic cells via imaging flow cytometry addresses my initial comment, and I hope that the authors agree that the new data in Figure 7 help strengthen their manuscript. One minor thing that I think would benefit the manuscript would be a supplementary figure panel showing some additional control samples with the gating strategy used to differentiate between large nuclei (necroptotic) and small nuclei (apoptotic); it would be good to show where live cells fall within these gates to ensure that they are not being counted as either PCD population.

(2a-c)The Western blot analysis of phospho-MLKL, phospho-RIPK1 and/or phospho-RIPK3 (Fig. 7e-g) helps strengthen the author's statements regarding necroptosis induction in these cells following combination treatment. The application of the much more specific RIPK3 inhibitor GSK872 (Fig. S7e) is also helpful.

(3) I understand that trying to update a computational model to incorporate primary human tumor proteomics data as well as XIAP expression would expand the computational modeling aspect of this project beyond the scope of this paper. I hope that the authors do explore this in the future as I think it would be an interesting follow-up study to the work presented here. I'm glad that the authors found the Schmid (2013) citation helpful for their discussion as well.

Reviewer #3

(Remarks to the Author)

The authors addressed majority of my questions.

I still have two small questions for the authors to clarify.

1 In Supp Figure 3a of current version, the authors did not show any Cl-caspase-3 and continue making the conclusion that the low sensitivity of SUI-020 cells towards S63845 treatment was also observed in Western Blot analysis of caspase cleavage. Without including a positive control sample, it is difficult to judge that the undetectable caspase-3 cleavage was due to short exposure of the membrane. The author also did not show any Cl-caspase-3 under the same treatment of the other two cells lines in the manuscript. Therefore, I think the experiment design was not properly controlled to make a precise conclusion.

2 The variation of the figure 3a is still too big. Can the author show the repeated data of this figure? The non-significance of 10uM treatment seems to be due to big variation, therefore it might not reflect the real situation.

Point to point response

Reviewer #1 (Remarks to the Author):

In the manuscript “Pharmacological targeting of c-FLIP_L enhances cell death *via* increase of complex II assembly in pancreatic cancer cells”, Corinna König *et al.*, demonstrate synergy in targeting Death receptor pathways in combination with gemcitabine and Mcl-1 inhibitor.

I think the data is interesting and worth considering but need further improvements. I am particularly concerned and not convinced about the conclusions on the necroptosis driven by combining Trail/CD95 with Mcl-1inh and gemcitabine. Additionally, I would like to see more relevant 3D model system of pancreatic cancer/normal included in the study.

General answer

We thank the reviewer for taking the time to review the manuscript. We have performed additional work to address the reviewer's comments, and we hope that the reviewer will be satisfied with our responses.

To summarize our revisions, we would like to point out that in the revised version of the manuscript we have added:

- The experiments with the third pancreatic cell line Panc89, which reproduce all the main results obtained with SUIT-020 and MiaPaca2 cells
- The experiments with non-malignant human fibroblasts as a model of 'normal cells' that were largely resistant to these treatments
- The experiments with the 3D model of pancreatic cancer cells based on Panc89 cells that were very sensitive towards these treatments
- Further experiments to demonstrate that we observed the induction of necroptosis upon DL/gemcitabine/FLIPinB/Mcl-1 inhibitor treatment in **combination with zVAD-fmk**

In particular, to further support the induction of necroptosis upon DL/gemcitabine/FLIPinB/Mcl-1 inhibitor/zVAD-fmk treatment, we have added to the revised version of the manuscript:

- Imaging experiments showing the appearance of the cells with the typical necrotic or necroptotic morphology, the number of which increases upon zVAD-fmk addition (Figure 7a)
- Quantification of apoptotic versus necrotic/necroptotic populations (Figure 7b, c)
- Western Blot analysis of necroptosis markers, *e.g.* pRIPK1, pRIPK3 and pMLKL (Figure 7d-f)
- Necrosome assembly upon DL/gemcitabine/ FLIPinB/Mcl-1 inhibitor/zVAD-fmk treatment (Figure 9c, Supplementary Figure 10)
- Experiments with the RIPK3 inhibitor GSK872, which rescues the cells from DL/gemcitabine/FLIPinB/Mcl-1 inh/zVAD-fmk treatment (Supplementary Figure 7e)

Please, see our detailed answers below.

Major points:

1. **(a)** Conclusions in the paper are based on two pancreatic cancer cell lines and it would be good if the authors included a panel of pancreatic cell lines to see how general these findings are.

Answer 1a

We have added the experiments with the third cell pancreatic cancer cell line, Panc89. We repeated all key experiments with this cell line, which included cell viability assays, caspase activity assays and Western Blot analysis upon single, double, triple and quadrupole treatments. The findings, which were obtained in MiaPaca2 and SUIT-020 cells that quadrupole treatment sensitizes the cells, were reproduced in Panc89 cell line.

A very important observation is that Panc89 cells also had a different response to DLs, Mcl-1 inhibition and FLIPinB compared to MiaPaca2 and SUIT-020, which is most likely based on different expression levels of key cell death proteins as highlighted below. Nevertheless, this cell line undergone cell death in the most efficient way upon quadrupole treatment, further supporting the strength of our approach and the need to target different components of the cell death network to efficiently sensitize cancer cells to cell death.

(b) One of the cell lines SUIT-020 are resistant to death ligands, could that be explained? Is it something to do with death machinery (apoptotic/necroptotic) or it is result of the death receptor activation? This can be addressed by using positive control: treating the cells with TNF and Smac mimetic (for apoptosis) or TNF, Smac mimetic and zVADfmk (for necroptosis).

Answer 1b

Following the reviewers' comments, we performed Western Blot profiling of the expression of all key components of the extrinsic and intrinsic cell death network in three cell lines under study: SUIT-020 *versus* MiaPaca2 and Panc89 cells. This profiling included the key components of the death receptor pathway, mitochondria pro- and anti-apoptotic regulators and apoptotic inhibitors such as XIAP (Figure 1a-c). Importantly, SUIT-020 cells have the higher expression of XIAPs and Bcl-2 compared to the other cell lines. Please, see the new Figure 1a. We put this important point into the discussion as it supports our observations and shows that increase in caspase-8 activity induced by FLIPinB allows to overcome the apoptotic resistance caused by high levels of XIAP in these cells.

The expression of most of the core components in SUIT-020 cells is comparable to the other pancreatic cell lines. Only the expression of TRAIL-R2 seems to be lower in SUIT-020 cells compared to MiaPaca 2 cells (Supplementary Figure 1). However, DR4 and caspase-8 are expressed at the same level in all three cell lines and FADD is comparable between the three cell lines with the lowest level in MiaPaca2 cells (Figure 1a). The other puzzle is that CD95 has the highest expression in SUIT-020 cells among three cell lines and they are still very resistant to CD95 stimulation. However, this observation fits well to the high expression levels of XIAPs in this cell line and supports their resistance towards DL stimulation. We also did not observe higher expression levels of anti-apoptotic members of the BCL-2 family in SUIT-020 (Bcl-2, Bcl-Xl and Mcl-1).

Collectively, considering the high XIAP expression in SUIT-020 cells it is not surprising that these cells are more resistant to DL treatment. We added small text on this in the discussion as well as commented on it in several positions in the manuscript text. Please, see the text in the Discussion below (page 19, line 478):

Among these three cell lines, the SUIT-020 cells had the highest level of XIAP expression. Accordingly, they were less sensitive to DL administration as well as to FLIPinB administration, supporting the critical role of the XIAP/caspase-3 ratio for the promotion of

apoptosis initiated by caspase-8 activation^{30,31,44,45}. Thus, the enhancement of caspase-8 activity at the DISC by FLIPinB in SUIT-020 cells was apparently not sufficient to induce apoptosis due to the blocking effects of XIAPs downstream of caspase-8 activation. This further supports the efficiency of our approach and requirement to target simultaneously different core components of cell death network like it was done in our study using the combination with targeting Mcl-1.

The expression of the core components of the **necroptotic** network in pancreatic cancer cell lines was compared with colon cancer HT29 cells as the reviewer suggested. MiaPaca2 cells did not express RIPK3 while Panc89 cells showed the higher expression of RIPK1, RIPK3 and MLKL compared to SUIT-020 cells and Panc89 were also more sensitive to necroptosis. Hence, this analysis also explains the lower sensitivity of SUIT-020 compared to Panc89 cells towards necroptosis.

Regarding the induction of necroptosis in SUIT-020 cells by combining Trail/CD95L with Mcl-1 inh, gemcitabine and zVAD-fmk, we have added more experiments to this manuscript, in particular the analysis of pRIPK1 and pMLKL in this cell line, quantification of flow cytometry measurements as well as the analysis by cellular morphology. We hope that the reviewer will find these data convincing.

With respect to the sensitivity of SUIT-020 cells towards DR/BV6/zVAD-fmk treatment, please, see below the set of experimental data for the reviewer-only. These experiments demonstrate that necroptotic markers can be detected upon CD95L/BV6/zVAD-fmk treatment. In these experiments, we also use HT29 cells treated with CD95L/BV6/zVAD-fmk as positive control, which is a common model for necroptotic cell death as was pointed out by the reviewer. Furthermore, the administration of Nec-1s led to inhibition of the phosphorylation of RIPK1 and MLKL in both SUIT-020 and HT29 cells.

Figure 1 for the reviewer: SUIT-020 cells show appearance of necroptotic markers upon CD95L/BV6/zVAD-fmk stimulation. (A-D) HT29 cells (A,B) or SUIT-020 cells (C,D) were pretreated for 1 h with zVAD-fmk, BV6 and Nec-1s and then treated for the indicated time intervals with CD95L. Total lysates were analysed by Western Blot with the indicated antibodies.

2. Authors are stating: “In our study, we have aimed at selecting the lowest concentrations of the cell death inducing stimuli which when applied alone do not induce any cytotoxic effects but work in combination to eliminate cancer cells. In this way, we aimed to mimic the effects for reducing the possible side effects of combinatorial treatment using cell lines as a model system.”

Definitely, there is a logic to minimise toxicity by reducing the dose and combining with drugs that target different pathways, however there is no evidence in the work presented about the response of normal, non tumour cells. How selective this combination is for pancreatic cancer cells? One way would be to expand the study using pancreatic derived cancer organoids and compare them with normal organoids. Such model system will increase the strength of this study.

Answer 2

We thank the reviewer for this valuable comment. We used non-malignant **human fibroblasts** in the revised version of the manuscript as “a normal tissue”, which have shown **almost complete resistance** towards treatments used in our study. The data are presented in the Supplementary Figure 6a-d. This is a very important point and we are grateful that the reviewer asked for it. Please, see the new text, Results, line 344:

To test whether this combinatorial treatment induces cell death in non-cancerous cells, the normal human fibroblasts expressing CD95, TRAIL-R1 and TRAIL-R2 were used (Supplementary Figure 6a). Importantly, no increase in cell death was observed for normal human fibroblasts treated with TRAIL/gemcitabine/FLIPinB/S63845 indicating the higher efficiency of this combination for cancer cell lines (Supplementary Figure 6).

3. In figure 1 a-it is good to indicate each line what represent. It is described in the text however the figure should be clear.

Additionally, RIPK3 levels are not that conclusive. It is very important to draw conclusions since necroptosis is studied. Authors need to use a positive control for RIPK3 expression so we can have an idea what is the level of RIPK3. Such positive control is HT29 that has been studied extensively in the necroptosis field and I would like to see what is the relative level of RIPK3 expression between these cell lines. And as I suggested above, positive control of necroptosis using TNF+Smac mimetic+zVADfmk for SUIT-020 and if HT29 could be included as control. It will be important so we know how necroptosis works in SUIT-020.

Answer 3

The lines in these Western Blots (former Figure 1a) corresponded to different cell passages. We have provided a clear indication under each line, e.g. by indicating the cell passages under the lines in Western Blot. It has to be noted that the former figure 1 is currently in supplementary material, since in the Figure 1 we now show the comparison between three different cell lines, where we also indicate the passages directly in the figure.

RIPK3 levels: The new Figure 1a shows the expression of all major cell death proteins as well as that of RIPK3. In fact, the expression of RIPK3 is lower in SUIT-020 compared to Panc89 and HT29 cells, but still detectable. This is consistent with our observations that this cell line undergoes necroptosis, but less efficiently compared to Panc89 cells.

We have presented the data above (response 1b) for the reviewer to show that necroptosis "takes place" in SUIT-020 cells under the conditions used in this study as well as upon CD95L/BV6/zVAD-fmk stimulation. We hope that the response 1b addresses all of the reviewer's comments on the necroptosis response in SUIT-020 cells.

4. In figure 1 b-e, I would suggest siRNA against FADD be used to demonstrate specificity.

Answer 4

For the comment to this Figure, we fully agree with the reviewer that we do not observe the DL-mediated dose-dependent decrease of cell viability for SUIT-020 cells (Figure 1b, c), which asks for the additional controls like FADD siRNA. We do observe DL-mediated dose-dependent decrease of cell viability for Panc89 cells treated with TRAIL or CD95L (Figure 1f, g) as well as for MiaPaca2 cells treated with TRAIL. The latter indicates that in these two cell lines DR response is of higher strength compared to SUIT-020 cells. The observed effects might indicate that we use concentrations of DLs that are rather low or at the threshold level for SUIT-020 cells, which is highly likely due to the high expression levels of XIAPs in this cell line as we found out in the course of the revisions. As mentioned in the answer 1, we discuss the higher resistance of SUIT-020 cells towards DL stimulation, which we think is based on the high expression of XIAPs.

Nevertheless, we provide the arguments below to argue that there is FADD- and caspase-8-dependent activation in this cell line upon DL/gemcitabine treatment. For the FADD-dependent activation of DRs in SUIT-020 cells, corresponding to experiments in Figures 1b,c we have the following arguments:

- we observe c-FLIP processing upon DL and DL/gemcitabine treatment under the similar conditions as in figure 1 b-c (Supplementary Figure 2), which supports that this process is FADD-dependent as FADD is a major adaptor for caspase-8 at the DISC, leading to caspase-8 activation, which in turn cleaves c-FLIP.
- In the previously published manuscript by Pietkiewicz et al, we have shown DISC assembly after DL and DL/gemcitabine in SUIT-020 cells, where we see c-FLIP processing as well as FADD recruitment to the receptor, further supporting the activation of the CD95L pathway in SUIT-020 cells upon these treatments. We also present DISC experiments in the current manuscript in Supplementary Figure 8a. These experiments still demonstrate DISC assembly and caspase-8 activation in SUIT-020 cells.

Reference:

Pietkiewicz, S. et al. Combinatorial treatment of CD95L and gemcitabine in pancreatic cancer cells induces apoptotic and RIP1-mediated necroptotic cell death network. *Exp Cell Res* **339**, 1-9, doi:10.1016/j.yexcr.2015.10.005 (2015).

Please see the corresponding Figure on the DISC assembly from the

manuscript Pietkiewicz, S. et al. below:

Figure 2 for the reviewer: SUIT-020 cells were pretreated for 24 h with 100 ng/ml gemcitabine and afterwards treated with 333 ng/ml CD95L for 30 min. CD95DISC was immunoprecipitated using anti-APO-1 antibodies. IP was analysed using Western Blot with the corresponding antibodies. Abbreviation: IP Immunoprecipitation, b bead control

5. In suppl fig 2a-d-why there is no more cleaved casp8/casp3 when combination of gemcitabine and TRAIL is used? What is the evidence that casp8 activation drives this death?

Answer 5

We used a dose of both agents in these experiments that should lead to the relatively low response and, accordingly, low or threshold levels of caspase cleavage products. In addition, these cleavage products are not stable and are rapidly degraded by proteasome. Although we do see a very small increase in caspase-3-p19/p17 cleavage product as well as PARP1 upon TRAIL/gemcitabine co-treatment, we agree that these effects are rather at the threshold level. However, here we would like to underline that our goal in this study was to show that these stimulations alone induce apoptosis at

the threshold level or very little apoptosis but when combined they promote a stronger response. Subsequently, this figure is only to show what is happening to caspases and that their activation is indeed at the ‘threshold’ level.

We agree that the previous description of these experiments might have been misleading, as we had made the statement on increase of caspase cleavage under these conditions. Hence, we have modified the corresponding text. Please, see the new text, page 8, line 186:

DL/gemcitabine-induced cell viability loss upon co-treatment with 10 ng/ml of gemcitabine was not accompanied by an increase of procaspase-3 and PARP1 cleavage products as was observed in SUIT-020 cells (Supplementary Figure 2a-d). This pattern fits well to the previous observations of caspase processing upon the stimulation with threshold concentrations of DLs or other death stimuli ³³.

As mentioned in answer 4, we observe DISC assembly leading to caspase-8 activation upon CD95L/gemcitabine treatment, which supports activation of caspase-8 in this pathway.

6. Fig 3d-e. The effect of triple combination is not very convincing because double (DL/gemcitabine) can cause similar effect.

Answer 6

The experiments in Figure 3 are devoted to the selection of the concentration of Mcl-1 inhibitor for the quadrupole treatment. We have changed the text accordingly and apologize for not describing this properly in the previous version. The selection of the concentration of this inhibitor was aimed to take ‘the first concentration above the threshold’, *e.g.* the concentration that is inbetween ‘no-cell-death’ or just the cells starting to die. We added the corresponding remark to the manuscript text.

We would like to note, that the fact that triple treatment does not increase cell viability loss compared to double one, further supports the strength of our approach and the need to target different components of the cell death pathway, as by adding the fourth component, FLIPinB, we can increase the loss of cell viability in these cells.

As highlighted above, we have expanded the text to comment on the way we selected the concentration of Mcl-1 inhibitor. We thank the reviewer for this comment. Please,

see the new text below, page 10, line 240:

... Next, we examined the concentrations of S63845, which can be used for sensitization of these three cell lines towards DL/gemcitabine/FLIPinB treatment. For combinatorial treatments, we have selected low concentrations of gemcitabine and S63845, since as highlighted, our aim was to use low or threshold doses of stimuli that work in combination to eliminate the cancer cells. Treatment with 10 ng/ml gemcitabine was used for all cell lines as discussed above. To select the concentration of S63845, this inhibitor was added to the cells in a dose-dependent manner in combination with DL/gemcitabine. The threshold concentration of S63845 was then selected as the concentration in the range between the one that caused the decrease of cell viability and the one that did not influence the cell viability loss. Specifically, in SUIT-020 cells, 20 μ M S63845 in combination with DL/gemcitabine induced loss of cell viability, while 10 μ M did not (Figure 3d, Supplementary Figure 3b). Therefore, 10 μ M S63845 was chosen as the concentration for treatment of SUIT-020 cells (Figure 3d, Supplementary Figure 3b).

7. TNF is not classical NF κ B gene and is more regulated by p38. I would suggest the authors to use some of the following NF κ B genes: IL6/SOD2/BIRC3/A20.

Answer 7

We again have to apologize for not perfect explanation here. The aim of these experiments was to verify TNF α production, as it has also been reported to be increased by gemcitabine administration in the previous studies. May be this would have led us into NF κ B pathway. This needed to be evaluated together with the effects on the putative effects on DR expression upon gemcitabine treatment, which were also reported previously. Therefore, in the current version, we present the measurements of DR and TNF α expression following gemcitabine treatment. In the previous version of the manuscript, we described rationale for TNF production experiments as possible NF κ B activation, although we fully agree with the reviewer - this may be too imprecise and require more experiments. We have rewritten this part of the text accordingly; we no longer mention NF κ B activation here as a basis for these experiments but we support the rationale behind these experiments by possible

upregulation of TNF α upon gemcitabine treatment, which was reported before and has to be evaluated in our study. In addition, reviewer 3 commented that the TNF α value we obtain is negligible, so we have also changed our text accordingly. See the new text below, page 12, line 310:

The treatment of cancer cells with gemcitabine has been reported to induce upregulation of the DL and DR level, in particular, TRAIL-R2 and TNF α ³⁷⁻⁴⁰. This, in turn, can lead to an increase of cell viability loss upon DL/gemcitabine/FLIPinB/S63845. Accordingly, the changes in the expression of TRAIL-Rs upon double (DL/gemcitabine), triple (DL/gemcitabine/S63845) and quadrupole (DL/gemcitabine/FLIPinB/S63845) treatments were monitored. However, no upregulation of TRAIL-Rs upon DL/gemcitabine/FLIPinB/S63845 was observed compared to double and triple treatments (Supplementary Figure 5a). TNF- α production was slightly increased upon triple and quadrupole co-treatments, but the levels of TNF- α increase were rather low (Figure 5d). Therefore, we suggested that upregulation of TNF- α also did not contribute to the increase in cell viability loss. These experiments ruled out the involvement of DL/DR upregulation upon this co-stimulation and suggested further exploring the molecular network of DL/gemcitabine/FLIPinB/S63845 co-treatment.

8. Fig 5e- This is not accurate assay to distinguish between apoptosis and necroptosis. The issue is that there are many cells that are not 100% apoptotic or necroptotic and that will make very difficult to split the percentage of apoptosis versus necroptosis. The way to demonstrate whether the cells can die by necroptosis is by using different inhibitors like RIPK1 inh and caspase inh. And in figure 6 authors did use them. However I am not convinced that there is necroptosis since zVADfmk in presence of Trail/CD95 protects. If that was the case, It should sensitise to cell death-necroptosis. Additionally, I would suggest the authors to use more selective inhibitors of RIPK1 like GSK'963 or PK68 that work in nM concentrations rather Nec1S that is not very selective. Apart of cell death assay it will be important to show necrosome formation by immunoprecipitation and activation of RIPK3/MLKL on western blot.

Answer 8

We thank the reviewer for the valuable comments. We have added to the current version of the manuscript:

- Imaging experiments showing the appearance of cells with necrotic morphology in the cell population (Figure 7a). The addition of zVAD-fmk significantly increased the number of these swollen, "necrosis-morphology-like" cells. This suggests the possibility of necroptosis induction
- Western blot analysis of necroptosis induction, e.g. pRIPK1, pRIPK3 and pMLKL, showing the appearance of phosphorylated forms of these kinases (Figure 7d-f)
- Experiments with the RIPK3 inhibitor GSK872, which rescues the cells upon of DL/gemcitabine/FLIPinB/Mcl-1inh/zVAD-fmk treatment (Supplementary Figure 7e)
- Quantification of apoptotic versus necroptotic populations (Figure 7b,c)
- Necrosome analysis by immunoprecipitation from Panc89 cells, demonstrating the assembly of the activated necrosome

In combination with the data present in the first submission, we are convinced that now we provide more evidence for necroptosis induction. However, it has to be noted that we observe necroptosis induction upon addition of zVAD-fmk

As for the ATP assays presented in the former figure 6, they are now in the supplementary material. We fully agree with the mentioned above issues. Indeed, the mentioned above issues might come from the fact that ATP assays are rather indirect measurements of cell viability or nec1s is not so efficient in this system.

Reviewer #2 (Remarks to the Author):

Cancer cells are typically characterized by their ability to resist programmed cell death (PCD) pathways, promoting tumor survival. This is often achieved through loss or decreased expression of PCD signaling pathway components, such as enzymes required for classical apoptosis or alternative PCD forms such as necroptosis. Strategies to restore or enhance PCD pathways in tumor cells, including the design of novel therapeutics with fully characterized mechanisms of action, represent an important area of cancer research with clear translational relevance. This group had previously reported the design of a novel “FLIP interactor” molecule (FLIPinB) that functioned by binding to and stabilizing c-FLIPL, an important pro-apoptotic regulator of extrinsic apoptosis. Because c-FLIPL is a component of the death-inducing signaling complex (DISC) that binds to pro-caspase-8, this stabilizing interaction between the c-FLIPL and caspase-8 heterodimer promotes the cleavage of pro-caspase-8 into active caspase-8, resulting in extrinsic apoptosis induction following activation of death receptors such as CD95 or TRAIL. Despite these intriguing findings, the complete molecular mechanisms of FLIPinB activity, particularly in relevant cancer cell lines rather than HeLa cells, remained incompletely understood. Establishing such information about FLIPinB action would provide essential information needed to realistically assess the value in carrying this drug candidate forward as a novel therapy in future translational studies.

Here, Konig and colleagues expand upon their previous work by providing a detailed overview of FLIPinB’s molecular mechanism of action. Using 2 pancreatic ductal adenocarcinoma (PDAC) cell lines with differing death receptor (DR) ligand sensitivities, they show that combinatorial treatment of various death modulators with FLIPinB increases pancreatic tumor cell death via apoptotic and necroptotic signaling pathways, and that this is due to an increase in pro-death complex II assembly and activation.

This study is thorough and well-written, and addresses an interesting area of cancer research with potential to inform novel translational studies for pancreatic cancer.

General answer

We thank the reviewer for taking the time, checking the manuscript and providing very thoughtful comments. We did our best to address the comments of the reviewer and hope that the reviewer will be satisfied with our answers.

Comments

There are a few aspects of the study that I believe would benefit from further

experimental investigation and/or discussion in order to strengthen the authors' conclusions and overall impact of the manuscript. Here is a summary of my primary comments:

1. Showing specificity of death effects in pancreatic tumor cells vs. healthy cells

1a. The majority of experiments are performed in 2 PDAC cell lines in culture: MiaPaca2 as a model sensitive to extrinsic apoptosis via DR ligands, and SUIT-020 as a DR ligand-resistant model. However, it is unclear whether the combination therapies tested also have cytotoxic effects on non-cancerous cells, including healthy pancreatic cells.

Answer 1a

We thank the reviewer for these comments. We have performed experiments with non-cancerous cells, namely using human fibroblasts, which were largely resistant to these treatments. Please see the new Supplementary Figure 6 as well as text below. This further confirms the efficiency of our strategy. Please, see the new text, Results, line 344:

To test whether this combinatorial treatment induces cell death in non-cancerous cells, the normal human fibroblasts expressing CD95, TRAIL-R1 and TRAIL-R2 were used (Supplementary Figure 6a). Importantly, no increase in cell death was observed for normal human fibroblasts treated with TRAIL/gemcitabine/FLIPinB/S63845 indicating the higher efficiency of this combination for cancer cell lines (Supplementary Figure 6).

It should be added that at the suggestion of another reviewer, we have included a third cell line in this manuscript, Panc89, which also serves as another model for sensitive pancreatic cancer cells. We repeated all key experiments with this cell line, including cell viability assays after single, double, triple and quadrupole treatments, caspase activity assays, Western Blot analysis and immunoprecipitations. Results obtained in MiaPaca2 and SUIT-020 cells can be reproduced in this cell line. In this way, we have a broader comparison of the effects of our treatment on different types of pancreatic cancer cells and show that treating pancreatic cancer cells with different levels of expression of key apoptotic proteins is very efficient by targeting several nodes of extrinsic and intrinsic cell death networks.

1b The authors shouldn't be asked to repeat all of these experiments in non-cancerous

cells, as this represents a considerable amount of extra work. However, it would be helpful to see cell viability data for either (a) a healthy pancreatic cell line (such as HPDE6c7, Millipore #SCC442) or (b) primary human pancreatic epithelial cells (such as Accegen #ABC-TC3744) treated with gemcitabine + death receptor ligand (CD95L or TRAIL), +/- FLIPinB. If these combinatorial treatments specifically lead to tumor cell death (as seen in MiaPaca2 cells in Fig. 2E) with minimal observed cytotoxicity in healthy cells, this would considerably strengthen the translational value of the manuscript by allowing claims regarding the potential safety of combination therapies in future translational studies. If the combination therapies induce equivalent amounts of cell death in non-transformed cells, this should be included as a potential source of off-tumor toxicity in translational studies.

Answer 1b

As highlighted above, we have performed experiments in human fibroblasts using them as a model of healthy pancreatic cells. Please, see the new supplementary Figure 6. Interestingly, we did not see toxicity for all combinations that exclude MCL-1 inhibitor.

2. Points related to programmed cell death pathways in PDAC cell lines treated with combination therapy

2a To characterize the mechanism of programmed cell death (PCD) by which SUIT-020 cells are dying following combination treatment, the authors first show morphological changes + PI uptake via AMNIS. The sample images shown in Fig. 5E look convincing with respect to necroptotic and late apoptotic cell identification, but it would be helpful to provide quantification of what percentage of cells analyzed fall into each category (viable vs. necroptotic vs. late apoptotic) to add information about cell death dynamics on a population level. It would also be helpful to include more information about the exact treatment conditions of these cells and the timepoint at which they were harvested for flow cytometry; do these correspond to Fig. 5C (gemcitabine + TRAIL + FLIPinB + S63845) or Fig. 5D (gemcitabine + CD95L + FLIPinB + S63845)?

Answer 2a

We thank the reviewer for a number of great suggestions and remarks. We used the imaging flow cytometry approach to quantify the percentage of apoptotic *versus* necrotic/necroptotic cells as the reviewer suggested. Specifically, we selected the PI positive cells and assigned the size/the shape of the nuclei in these cells to the typical morphology in apoptotic or necrotic cells. From this analysis, we gated the cells with the shrunken nucleus as a population of apoptotic cells and the cells with the swollen

or enlarged nucleus as necrotic cells. Please, see the new Figure 7b, c. This allowed us to get the estimation of the amount of cells undergoing apoptosis *versus* necroptosis upon addition of zVAD-fmk.

We have added more indications for the figures to show exactly the conditions for which cells were collected for Imaging Flow Cytometry. Please, see the new Figure 6.

To further support necroptosis induction upon DL/gemcitabine/FLIPinB/Mcl-1 inhibitor/zVAD-fmk treatment, we performed the following experiments and added them to the manuscript:

- Imaging experiments showing the appearance of cells with necrotic morphology in the cell population using SUIT-020 and Panc89 cells (Figure 7a). The addition of zVAD-fmk significantly increased the number of these swollen, "necrotic morphology-like" cells. This provides the evidence that we have both apoptotic and necroptotic cells after this treatment.
- Quantification of apoptotic versus necroptotic populations (Figure 7b,c)
- Western Blot analysis of necroptosis induction, *e.g.* pRIPK1, pRIPK3 and pMLKL, showing the appearance of phosphorylated forms of these kinases (Figure 7e-g).
- Experiments with the RIPK3 inhibitor GSK872, which rescued the cells from DL/gemcitabine/FLIPinB /MCl-1 inhibitor/zVAD treatment (Supplementary Figure 7e).
- Necrosome analysis by immunoprecipitation, demonstrating the assembly of the activated necrosome.

We are confident that this further support the induction of necroptosis in our system.

2b The results section for Figure 6A/B states that SUIT-020 cells die through a combination of necroptosis and apoptosis, using a set of experiments using stimulation with either TRAIL (Fig. 6A) or CD95L (Fig. 6B) with combination therapy +/- the RIPK1 inhibitor necrostatin-1s (Nec-1) or the pan-caspase inhibitor zVAD-fmk. Although the authors state that Nec-1s or zVAD-fmk administration alone leads to a "partial rescue" of SUIT-020 viability, these differences (white bars in 6A and 6B, 3rd and 4th sets of bar graphs) are not statistically significant and should be discussed as equivalent amounts of cell viability. Only combination treatment with Nec-1s + zVAD-fmk significantly restores cell viability to that observed with DMSO vehicle treatment (white bars in the 1st set of bar graphs).

Answer 2b

We agree with the comments of the reviewer on ATP assays that were presented in the previous version of the manuscript, it might be that nec-1 was not efficient in this system. Therefore, in the current version of the manuscript we show more evidence for the necroptosis induction using the other approaches as highlighted above in the Answer 2a. This particular experiment of measurement ATP production in the current version of the manuscript we present in the shortened version in the supplementary figure 8.

2c Nec-1s is used in Fig. 6, but this drug inhibits RIPK1; while RIPK1 activation is involved in necroptosis, it can also serve death-independent signaling functions if RIPK3 and MLKL are not activated downstream of RIPK1 (reviewed by Newton K, Cold Spring Harb Perspect Biol 2020). Claims regarding necroptosis would therefore be strengthened by revision experiments providing additional molecular characterization of the activation of different necroptosis signaling components. This would be particularly helpful for SUIT-020 cells, which express both caspase-8 and RIPK3 (Fig. 1A) and should be capable of undergoing both forms of PCD. Additional information could include Western blots for phospho-RIPK3 and/or phospho-MLKL (only found in necroptotic cells) in wild-type SUIT-020 cells, or genetic tools to knock out or knock down necroptotic signaling machinery in SUIT-020 cells and seeing how their deficiency influences cell viability. These types of tools could be used to answer the following questions:

Does combination therapy (gemcitabine + TRAIL/CD95L + FLIPinB + S63845) lead to an increase in phospho-RIPK3/phospho-MLKL in wild-type cells? Is the presence of these phosphorylated necroptosis pathway components blocked following administration of Nec-1s? Does removal of necroptotic signaling components influence SUIT-020 viability following combination therapy administration (gemcitabine + TRAIL/CD95L + FLIPinB + S63845)? Such manipulations could include generating RIPK3^{-/-} or MLKL^{-/-} SUIT-020 cells using CRISPR/Cas9, or knocking down gene expression using stable transduction of shRNA/transient transfection with siRNA.

Answer 2c

We have performed the Western Blot detection for pRIPK1, pRIPK3 and pMLKL. Please, see this analysis below and in the new Figure 7. The detection of pRIPK3 was not optimal due to the low quality of antibodies that we have, however, pMLKL and pRIPK1 were detected reasonably well indicating necroptosis induction. We also performed the corresponding analysis in Panc89 cells, showing the appearance of pRIPK1 and pMLKL in these treatments.

Figure 3: SUIT-020 cells were pretreated for 24 h with gemcitabine, 2 h with FLIPinB and S63845, 1 h zVAD-fmk and afterwards with CD95L for 5h. Western Blot was analysis was performed using the corresponding antibodies.

In addition, we used the RIPK3 inhibitor GSK872 which rescues the cell viability after gemcitabine/DL/S63845/FLIPinB/zVAD-fmk treatment. Please see the new Supplementary Figure 7e.

3. Modeling caspase-8:c-FLIP ratio in PDAC samples beyond MiaPaca2 and SUIT-020 lines

o The authors cite a previously published mathematical model (Ivanisenko & Lavrik, Biochemistry 2020) that they use to predict the potentially efficacy of FLIPinB treatment, based on the ratio of c-FLIPL to pro-caspase-8 expression in target cells. While they show where MiaPaca2 and SUIT-020 cell lines fall according to their model (Fig. 2B), it is unknown how these results compare to more clinically relevant PDAC patient samples. Since FLIPinB efficacy requires an optimal ratio of 2:1-3:1 pro-caspase-8/c-FLIPL and therefore may not work in a context of high pro-caspase-8/low c-FLIPL content (i.e. settings where there are lower amounts of pro-caspase-8/c-FLIPL heterodimers for FLIPinB to stabilize), it could be helpful to provide data showing that the ratios observed in MiaPaca2 and SUIT-020 cell lines fall within a similar range to that observed in primary patient samples.

o In the absence of access to primary human PDAC samples, if this group's model can use proteomics data as an input, they may consider analyzing published proteomics data (such as Cao et al, Cell 2021) to see where primary human PDAC cells (and healthy adjacent tissues) fall according to their model.

o This represents a fair amount of computational work and I do not know if the group's model is even compatible with the data outputs available from published proteomics data sets. However, since the likely efficacy of FLIPinB depends on the ratio of pro-caspase-8 and c-FLIP found in tumor cells, if it is possible to characterize human PDAC samples using this model then this may provide helpful data lending strong justification for pursuing FLIPinB combinatorial therapy in future translational studies.

o In the absence of any additional computational work, the authors may consider including a brief discussion of the work presented in Schmid SJ et al. (Pancreas 2013) showing that a majority of PDAC surgical resections were c-FLIP positive. Although this does not incorporate caspase-8 expression data, the fact that the majority of PDAC patients likely maintain intact c-FLIP expression could be promising when considering potential patient responsiveness to FLIPinB combination therapy.

Answer 3

We would like to thank the reviewer for giving the professional view on our mathematical modeling part and highlighting the valuable issues to address and how to improve the model. In the initial revision process, we tried to validate our model by the public databases following the very valuable suggestions made by the reviewer. In this process, we came across rather high variations in the databases, which resulted in the computationally very difficult task, which we were trying to accomplish.

However, at the later steps of the revision, we found out that an important role in our manuscript belongs to XIAPs that define the resistance of SUIT-020 cells. This asked for incorporation of XIAPs into the model as well as all downstream events and corresponding feedback loops. This would have made the model more complicated than simple ratios of caspase-8 to c-FLIP in different cell lines. This requires building of the expanded model, which is a subject of a future research. Hence, in the current version of the manuscript we omitted the modeling part and deleted the panel devoted to the mathematical modeling from the manuscript as well as the corresponding text. We discuss the ratios of c-FLIP to caspase-8 in the manuscript text and how it has the impact on the efficiency of FLIPin action citing our previous manuscripts, however, only in the form of discussion.

Furthermore, we are very much pleased with the beautiful suggestion to cite Schmid et al., 2013 and discuss the FLIP expression levels in pancreatic cancer cells as well as the impact of this information on our study. We have added this valuable point to the introduction and discussion and we are very grateful for this suggestion. Please, see the new text:

Introduction, line 121:

Finally, there is evidence that c-FLIP proteins may represent a promising target in PDAC, suggesting the importance of investigating pharmacological targeting of c-FLIP in this type of cancer ²⁹.

Discussion, line 462:

FLIPinB is the first-in class chemical compound targeting c-FLIP_L which has been recently developed and tested in HeLa, Jurkat and AML cell lines¹². Previous studies have shown the major role for c-FLIP in PDAC suggesting testing the effects of targeting c-FLIP in this type of cancer²⁹. The potential of FLIPinB in the elimination of pancreatic cancer cells in combinatorial treatments was explored in this study using three different pancreatic cancer cell lines.

In addition to the primary comments listed above, I also have a handful of minor criticisms that I believe would strengthen the manuscript if addressed by the authors:

1. Model figure readability

o The model figure presented in Fig. 8B is comprehensive but very busy. The authors may consider revising the model to remove several signaling components that are not directly examined in this manuscript (i.e. remove BID-tBID, XIAP, caspase-9, SMAC/DIABLO, MOMP, cytochrome-c), which may help simplify and highlight the key findings of the paper.

Answer 1. We have simplified the model but we left XIAP, as it turned out to play an important role in our study. We also left Bid- tBid to have some illustration of Mcl-1 action. Please, see the new Figure 10.

2. Polishing figure labels

o There are a few instances where figures could be updated be labeled more consistently. These include:

a) Fig. 3A, 3B: The differing font sizes in the table row for S63845 are distracting; all font sizes could be updated to be the same smaller size.

b) Fig. 3C: The concentrations for S63845 are written using “,” comma notation rather than using “.” decimal notation. To be more consistent with concentrations listed in other paper figures, please update to decimal notation.

Answer 2. We made changed the fonts to the same sizes. We changed comma notation to the decimal notation. We thank reviewer for these important comments.

Taken together, we would like to thank the reviewer for making the very important and thoughtful comments that allowed to improve the quality of our manuscript.

Reviewer #3 (Remarks to the Author):

Corinna König et al. described here that combined treatment with DL, gemcitabine, MCL-1 inhibitor S63845 and FLIPinB could efficiently decrease pancreatic cancer cell viability with relatively low dosages by inducing both apoptosis and necroptosis. While this is potentially interesting as a therapeutic strategy to treat pancreatic cancer patients, there are several main issues to be further clarified or evaluated. Moreover, the mechanism of this treatment-induced cell death needs to be further investigated. I therefore have the following comments that need to be addressed.

Major comments:

1. This study only tested the treatment in two representative pancreatic cancer cell lines MiaPaca2 and SUIT-020 cells. Since all the drugs used here are somewhat toxic to the cells, the authors need to show whether this treatment also affect the healthy normal cells in addition to cancer cells.

Answer 1: We have performed experiments with non-cancerous cells, such as human fibroblasts, which were resistant to these treatments, except for a small sensitivity to Mcl-1 inhibitor administration. Please see the new Supplementary Figure 6 as well as text below. This further confirms the efficiency of our strategy. Please, also see the new text, Results, line 344:

To test whether this combinatorial treatment induces cell death in non-cancerous cells, the normal human fibroblasts expressing CD95, TRAIL-R1 and TRAIL-R2 were used (Supplementary Figure 6a). Importantly, no increase in cell death was observed for normal human fibroblasts treated with TRAIL/gemcitabine/FLIPinB/S63845 indicating the higher efficiency of this combination for cancer cell lines (Supplementary Figure 6).

We have also included a third cell line in this manuscript, Panc89, which also serves as another model for sensitive pancreatic cancer cells towards both apoptosis and necroptosis. We repeated all key experiments with this cell line, including cell viability assays after single, double, triple and quadrupole treatments, caspase activity assays, western blot analysis and immunoprecipitations. The results obtained in MiaPaca2 and SUIT-020 cells can be reproduced in this cell line. Specifically, quadrupole treatment was very efficient in this cell line as well. In this way, we have a broader comparison of the effects of our treatment on different types of pancreatic

cancer cells.

2. I am not sure whether it is realistic to apply four toxic drugs in vivo to treat cancer patients. Particularly, it is important to minimize the side effects of these drugs, therefore, even if this is possible, the concentrations of the drugs need to be carefully tested and optimized again in vivo. Therefore, I would believe that performing in vivo experiments to prove that the treatment works in vivo would be necessary.

Answer 2: We tested the combined treatment in 3D cell culture based on Panc89 cells. We observed cell viability loss upon DL/gemcitabine/S63845/FLIPinB quadrupole treatment in 3D culture, but not with double or triple treatment. These experiments indicate that further development of this approach is promising. Please see the new Figure 5a-c as well as text below. Please, also see the new text, Results, line 280:

Importantly, the increase in cell viability loss upon quadrupole DL/gemcitabine/S63845/FLIPinB stimulation compared to double (DL/gemcitabine) or triple (DL/gemcitabine/S63845 and DL/gemcitabine/FLIPin) treatments was also observed in Panc89 cells growing in 3D as tumor-mimicking spheroids (Figures 5a-c). Importantly, in these experiments, the quadrupole treatment was the most efficient one for both CD95L and TRAIL treatments.

3. The FLIPinB affects extrinsic apoptosis pathway, while the MCL-1 inhibitor affects intrinsic apoptosis pathway, combined treatment of these two drugs with DL/gemcitabine therefore would affect both intrinsic apoptosis and extrinsic apoptosis. Thus, the authors should examine Caspase-9 cleavage as well. To more precisely address the mechanism, the authors should perform the experiments in Caspase-8 and Caspase-9 KO cells. Moreover, the authors only showed the happening of necroptosis by inhibiting RIPK1 kinase activity without showing any necroptosis markers activation, such as p-RIPK1, p-RIPK3 and p-MLKL in the lysates. The author should at least examine the necroptosis markers activation. Plus, I am not sure whether the very low (3-8pg/mL) TNF production presented in Figure 4d is meaningful or not.

Answer 3: We have performed the additional analysis of signaling in our system

following the valuable comments of the reviewer. Especially, the major extension of our study involved the analysis of necroptotic signaling. In particular, we investigated the necroptosis markers by Western Blot analysis and could observe the phosphorylation of RIPK1 and MLKL after quadrupole treatment in SUIT-020 and Panc89 cells. In addition to our imaging flow cytometry measurements we also checked the images of dying cells after DL/gemcitabine/S63845/FLIPinB plus minus zVAD treatment showing the typical apoptotic as well as necroptotic morphologies (Please, see the new Figure 7a). Furthermore, we performed experiments with the RIPK3 inhibitor GSK872, which rescued the cells from DL/gemcitabine/FLIPinB /MCI-1 inhibitor/zVAD treatment (Supplementary Figure 7e). Finally, we carried out necrosome analysis by immunoprecipitation from Panc89 cells, demonstrating the assembly of the activated necrosome (Figure 9).

We thank the reviewer for the comment about TNF α production, we fully agree with it and we changed the text accordingly. Now we discuss that there are minor effects on TNF α production that cannot have a contribution to the increase of cell death upon quadrupole treatment.

For experiments with caspase-9: in the current version of the manuscript we found that the key role in the cell death sensitivity of these cells belong to XIAPs that play a prominent role in our study. Looks like XIAP levels largely define Type I/Type II phenotype of pancreatic cells under study, hence we focus our discussions around XIAP and its role.

4. The effect of majority of the data are marginal and not obvious in my opinion. For example, in Figure 7a, Cl-Caspase-3 and Cl-Caspase-8 in TGFSF-treated samples are not obviously higher than TSF-treated samples; in Figure 7b, Cl-Caspase-3 in CGSF-treated sample is even lower than CG-treated sample; Honestly, I could not tell the difference between necroptosis and late apoptosis from the pictures presented in Figure 5e; in Supp-Figure 2, the difference of Cl-Caspase-3 and Cl-Caspase-8 are not obvious, even though the p17/19 of Cl-Caspase-3 are increased in TGFSF- or CGSF-treated samples, gemcitabine treatment alone already induced similar amount of Cl-Caspase-3 activation; in Supp-Figure 7a, the DISC components recruited to the complex are less in CGSF-treated samples than CGF-treated samples and similar with CSF-treated samples, which suggests that FLIPinB does not further increased DISC formation; the effect in Supp-Figure 8b is marginal.

Answer 4: We thank the reviewer for the critical comments and we would like to comment on each point below:

- *Comment to the Former Figure 7a/7b, Supplementary Figure 2 and the low*

level of caspase processing detected by Western Blot:

We use the low or threshold concentrations of the drugs inducing cell death, therefore, the effects are not so strong at the Western Blot level. This is particular the case for SUIT-020 cells that are very resistant to the cell death and were shown in the former supplementary Figure 2.

However, here we would like to underline that our goal in this study was to show that these stimulations alone induce apoptosis at the threshold level or very little apoptosis but when combined they promote a stronger response. Subsequently, this figure is only to show what is happening to caspases and that their activation is indeed at the ‘threshold’ level.

We rephrased the corresponding description in the text, corresponding to supplementary figure 2, to make this point clearer. In the previous version we stated that there are differences, however in the current version we state that the pattern is in accordance with the threshold level of stimulation. Hence, we have modified the corresponding text. Please, see the new text, page 8, line186:

DL/gemcitabine-induced cell viability loss upon co-treatment with 10 ng/ml of gemcitabine was not accompanied by an increase of procaspase-3 and PARP1 cleavage products as was observed in SUIT-020 cells (Supplementary Figure 2a-d). This pattern fits well to the previous observations of caspase processing upon the stimulation with threshold concentrations of DLs or other death stimuli ³³.

We used a dose of all agents in these experiments that lead to the relatively low response and, accordingly, low or threshold levels of caspase cleavage products. In addition, these cleavage products are not stable and are rapidly degraded by proteosome. Although we do see a small increase in caspase-3-p19/p17 cleavage product as well as PARP1 upon co-treatments, we agree that these effects are rather minor and should not be mentioned as an effect. As it was highlighted above, we would like to underline that our goal in this manuscript was to show that these stimulations alone or in double combination do not do anything or do apoptotic induction at the threshold level but when combined they promote a stronger response.

However, we agree that the previous description of these experiments might have been misleading and we did not stress the ‘threshold’ point enough, hence, we have modified the corresponding text as mentioned above.

As for the Figures 7a and 7b- now Figures 8a-c, we see some modest effects in SUIT-

020 and MiaPaca2 cells, while we see much stronger effects in the third line under study Panc89 cells.

- Comment to the Former Figure 5e and apoptotic/necroptotic images

As already stated before we also included now pictures of apoptotic and necroptotic cells with the characteristic morphology (Figure 7a) and hope that these pictures provide the additional confirmation of the necroptotic morphology. We are confident that these images show the detection of both apoptotic and necroptotic types of cell death. Furthermore, we largely increased the ‘necroptotic part’ of our manuscript. As for imaging flow cytometry images we provided more description in the text and we hope that this will be sufficient. Please, see the new Figure 7 and the new text below, line 350:

Imaging Flow Cytometry also allows distinguishing between apoptotic and necrotic or necroptotic cells using its imaging functions ⁴¹. Importantly, when analyzing PI-positive cells, along with late apoptotic cells that are characterized by shrunken nuclei, which can be observed as ‘sharp red spots’ (Figure 6e, f, right panels), a small number of cells with the typical morphology of necrosis or necroptosis were detected in both cell lines (Figure 6e, f, middle panels). In particular, the necroptotic or necrotic features include the swollen nucleus and an increase in cell volume, as observed by a more diffuse red color spread over the whole cell.

The appearance of cells with both apoptotic and necrotic morphology upon DL/gemcitabine/FLIPinB/S63845 treatment was also observed under the microscope, though the population of the necrotic cells was rather low (Figure 7a, middle panel). The addition of pan-caspase inhibitor zVAD-fmk increased the amount of cells with necrotic morphology in both SUI-020 and Panc89 cells indicating necroptosis induction upon zVAD-fmk addition (Figure 7a, right panel). Furthermore, we quantified the cells with necrotic morphology from imaging flow cytometry experiments (Figure 7b, c; Supplementary Figure

7a-d). For this quantification, we used an approach that was developed previously and is based on gating based on the size of the nucleus of apoptotic *versus* necrotic cells, e.g. distinguishing between small *versus* large nuclei using special features of imaging flow cytometry⁴¹. This analysis has shown that the necroptotic cells comprise about 60% of population upon DL/gemcitabine/FLIPinB/S63845/zVAD-fmk treatment (Figure 7b). This demonstrates that cells undergo necroptosis upon DL/gemcitabine/FLIPinB/S63845 treatment in combination with zVAD-fmk administration.

- *Comment to 'no increase' in the DISC assembly*

We did not claim that we have an increase in DISC assembly in the previous version. Importantly, we have less CD95 signal detected by Western Blot in the co-IP corresponding to quadrupole treatment indicating less efficient immunoprecipitation or loading and therefore the comments of the reviewer are well taken –it is less but due to loading. We also more offensively state that we see **no increase in the DISC formation** after DL/gemcitabine/S63845/FLIPinB treatment in the current version of the manuscript. DISC IP fractionated by size exclusion chromatography demonstrates only small effects on FADD recruitment for DL/gemcitabine/S63845/FLIPinB treatment further supporting our hypothesis that we have to test downstream events, e.g. formation of the complex II.

We remain with our main hypothesis that the quadrupole treatment promotes complex II assembly as was suggested previously and where we see an increase in the amount of all core components more clearly.

Minor comments:

1 Which groups are compared for statistical analysis in Figure 1? Control groups with all the other groups by paired-comparison with the same p-value or one p-value by one-way ANOVA is labelled on the graph?

Answer: We expanded the figure legend and hope that we could clarify this point. In particular, **line 863:**

For the statistical analysis One-way ANOVA tests were used to compare the group of conditions for each of TRAIL or CD95L treatments. Each group comprised the treatment with

one concentration of TRAIL/CD95L. The following values were used: **** $p < 0.0001$; *** $p < 0.001$; ** $p < 0.01$; * $p < 0.05$; ns not significant.

2 The variation in some experiments are too big to draw a precise conclusion. For example, in Figure 3a, the experiment need to be redo to draw a conclusion that S63854 does not induce cell death in SUIT-020 cells.

Answer: Here we again did not clearly explained our study in the previous version of the manuscript, which might have caused a confusion. The experiments in Figure 3 are only the selection of the concentration of Mcl-1 inhibitor for the quadrupole treatment. We have changed the text accordingly and apologize for not describing this properly in the previous version. The selection of the concentration of this inhibitor was to take ‘the first concentration above the threshold’, *e.g.* the concentration that is inbetween ‘no-cell-death’ or just the cells starting to die. We added the corresponding remark to the manuscript text.

In other experiments presented in different panels of this manuscript using the 10 μM concentration we could observe that the SUIT-020 cells are resistant towards this concentration when treated alone with S63845. Therefore, the results presented in 3a seem to be correct and 10 μM concentration is the concentration that does not cause a loss of cell viability in SUIT-020 cells. Please, see the new text below, page 10, line 240:

... Next, we examined the concentrations of S63845, which can be used for sensitization of these three cell lines towards DL/gemcitabine/FLIPinB treatment. For combinatorial treatments, we have selected low concentrations of gemcitabine and S63845, since as highlighted, our aim was to use low or threshold doses of stimuli that work in combination to eliminate the cancer cells. Treatment with 10 ng/ml gemcitabine was used for all cell lines as discussed above. To select the concentration of S63845, this inhibitor was added to the cells in a dose-dependent manner in combination with DL/gemcitabine. The threshold concentration of S63845 was then selected as the concentration in the range between the one that caused the decrease of cell viability and the one that did not influence the cell viability loss. Specifically, in SUIT-020 cells, 20 μM S63845 in combination with

DL/gemcitabine induced loss of cell viability, while 10 μ M did not (Figure 3d, Supplementary Figure 3b). Therefore, 10 μ M S63845 was chosen as the concentration for treatment of SUIT-020 cells (Figure 3d, Supplementary Figure 3b).

3 In Figure 3C, the authors should compare Cl-Caspase-3 activation between MiaPaca2 and SUIT-020 cells and include a positive control (Cl-Caspase-3 positive lysates) to make a conclusion that the low sensitivity of SUIT-020 cells towards S63845 treatment was also observed in Western Blot analysis of caspase cleavage.

Answer: We fully agree with these suggestions by the reviewer, however, since we are using different concentrations of Mcl-1 inhibitor in SUIT-020, MiaPaca2 and Panc89 cell lines, it would be difficult to carry on the corresponding comparison between these three cell lines.

However, we fully agree with the reviewer that it is not very informative when talking about one cell line. Hence, following the comment of the reviewer we moved this experiment to the supplementary material.

Point to Point Response

Reviewer #1 (Remarks to the Author):

I checked the response of the authors to my comments and I am happy that they included my suggestions and addressed my concerns. So I am happy for the paper to be accepted.

Answer

We thank the reviewer for the comments on our manuscript and for the time spent in carefully reviewing it.

Reviewer #2 (Remarks to the Author):

Overall I am impressed by the amount of work put into the revision of this manuscript and I feel that the authors put strong effort into addressing each of my initial critiques/suggestions.

I have listed any comments to individual points below:

(1) Specificity of death effects in PDAC cells versus healthy cells. The authors repeated numerous assays in human fibroblasts and show minimal toxicity in non-cancerous cells. They have also included a third PDAC line (Panc89) in their panel of key experiments which helps further strengthen their conclusions.

Answer 1

We thank the reviewer for carefully checking our results.

(2) Further elucidation of PCD pathways engaged in PDAC cell lines.

(2a) The quantification of apoptotic versus necroptotic cells via imaging flow cytometry addresses my initial comment, and I hope that the authors agree that the new data in Figure 7 help strengthen their manuscript. One minor thing that I think would benefit the manuscript would be a supplementary figure panel showing some additional control samples with the gating strategy used to differentiate between large nuclei (necroptotic) and small nuclei (apoptotic); it would be good to

show where live cells fall within these gates to ensure that they are not being counted as either PCD population.

Answer 2a

We thank the reviewer for this important remark and present the gating strategy in the new version of the manuscript. Please, see the new supplementary figure 7, specifically panels e, f, which are also shown below.

From Supplementary Figure 7 e, f: Gating strategy: the population of cells was separated in PI-negative (viable) and PI-positive (small and large nuclei, dead cells) in the dot plot

(2a-c) The Western blot analysis of phospho-MLKL, phospho-RIPK1 and/or phospho-RIPK3 (Fig. 7e-g) helps strengthen the author's statements regarding necroptosis induction in these cells following combination treatment. The application of the much more specific RIPK3 inhibitor GSK872 (Fig. S7e) is also helpful.

(3) I understand that trying to update a computational model to incorporate primary human tumor proteomics data as well as XIAP expression would expand the computational modeling aspect of this project beyond the scope of this paper. I hope that the authors do explore this in the future as I think it would be an interesting follow-up study to the work presented here. I'm glad that the authors found the Schmid (2013) citation helpful for their discussion as well.

General Answer

We thank the reviewer for the appreciation of our work and a number of very important comments that allowed to significantly improve the quality of our manuscript.

Reviewer #3 (Remarks to the Author):

The authors addressed majority of my questions.

I still have two small questions for the authors to clarify.

Comment 1

In Supp Figure 3a of current version, the authors did not show any Cl-caspase-3 and continue making the conclusion that the low sensitivity of SUIT-020 cells towards S63845 treatment was also observed in Western Blot analysis of caspase cleavage. Without including a positive control sample, it is difficult to judge that the undetectable caspase-3 cleavage was due to short exposure of the membrane. The author also did not show any Cl-caspase-3 under the same treatment of the other two cells lines in the manuscript. Therefore, I think the experiment design was not properly controlled to make a precise conclusion.

Answer 1

We have performed these experiments in three cell lines: SUIT-020, MiaPaca 2 and Panc89. This analysis has shown that SUIT-020 are much more resistant to Mcl1 inhibitor treatment compared to two other cell lines. Indeed, while in the other two cell lines the caspase cleavage was clearly detected, this was not the case in SUIT-020 cells. We also put SUIT-020 cells stimulated with CD95L on the same gel, which was used as a positive control as these stimulations resulted in the high level of caspase cleavage. Please, see the new supplementary Figure 3a-c as well as the updated text below. We fully agree that these experiments were very important.

New text, line 233:

The low sensitivity of SUIT-020 cells towards S63845 treatment was also observed in Western Blot analysis of caspase cleavage (Supplementary Figure 3a). Indeed, neither processing of procaspases-8, -10 and -3, nor cleavage of their substrates c-FLIP or PARP1 were detected upon treatment with up to 20 μ M of S63845 for 6 hours. This was in contrast to MiaPaca2 and Panc89 cells, in which procaspase-8 and -3 processing and cleavage of the caspase-3 substrate PARP1 were observed using even the lower concentrations of S63845 (Supplementary Figure 3b, c).

From Supplementary Figure 3a-c: Pancreatic cancer cells were treated with different concentrations of S63845 and with CD95L followed by subsequent Western Blot analysis.

Comment 2 The variation of the figure 3a is still too big. Can the author show the repeated data of this figure? The non-significance of 10uM treatment seems to be due to big variation.

Answer 2

We fully agree that these experiments were also very important and we have repeated them and added them to the manuscript. Please see the new figure 3a.

Figure 1a: Cell viability of SUI020 cells treated with the different concentrations of Mcl-1 inhibitor.